# The Multilateral Efficacy of Chitosan and *Trichoderma* on Sugar Beet

**DOI:** 10.3390/jof8020137

**Published:** 2022-01-29

**Authors:** Lisa Kappel, Nicole Kosa, Sabine Gruber

**Affiliations:** 1Department of Microbiology, University of Innsbruck, 6020 Innsbruck, Austria; lisa.kappel@uibk.ac.at; 2Department of Bioengineering, FH Campus Wien, University of Applied Sciences, 1190 Vienna, Austria; nicole.kosa@stud.fh-campuswien.ac.at

**Keywords:** *Trichoderma* *atroviride*, biocontrol, chitosan, crop control, plant pathogens, systemic resistance, plant defense, Cercospora leaf spot disease, *Beta vulgaris*, seed coating

## Abstract

The majority of all fungal formulations contain *Trichoderma* spp., making them effective biological control agents for agriculture. Chitosan, one of the most effective natural biopolymers, was also reported as a plant resistance enhancer and as a biocide against a variety of plant pathogens. An *in vitro* three-way interaction assay of *T. atroviride*, chitosan, and important plant pathogens (such as *Cercospora beticola* and *Fusarium oxysporum*) revealed a synergistic effect on fungistasis. Furthermore, chitosan coating on *Beta vulgaris* ssp. *vulgaris* seeds positively affected the onset and efficiency of germination. We show that priming with *T. atroviride* spores or chitosan leads to the induced expression of a pathogenesis-related gene (PR-3), but only supplementation of chitosan led to significant upregulation of phytoalexin synthesis (PAL) and oxidative stress-related genes (GST) as a defense response. Repeated foliar application of either agent promoted growth, triggered defense reactions, and reduced incidence of Cercospora leaf spot (CLS) disease in *B. vulgaris*. Our data suggest that both agents are excellent candidates to replace or assist common fungicides in use. Chitosan triggered the systemic resistance and had a biocidal effect, while *T. atroviride* mainly induced stress-related defense genes in *B. vulgaris*. We assume that both agents act synergistically across different signaling pathways, which could be of high relevance for their combinatorial and thus beneficial application on field.

## 1. Introduction

The bioactive molecule chitosan (CHSN) is on everyone’s lips and is considered one of the most effective natural antimicrobial agents. The activity of chitosan depends on its molecular weight (degree of polymerization (DP) was several 100 kDa to oligo chitosans, such as hexamers); the degree of deacetylation (DA), which is dependent on the pH (at pH 6.0 most amino groups are in free base form); and the target microorganism (decreasing efficiency: yeast > mold > Gram-positive bacteria > Gram-negative bacteria). Proposed mechanisms include membrane destabilization and host permeability (reviewed in [1]). Chito-oligosaccharides (*n* = 2–10) and very-low-molecular-weight chitosans (*n* ≤ 20) are mainly produced for therapeutic purposes, while the more viscous molecules (high molecular weight, HMW) are used for food and fiber technology [2]. Importantly, chitosan and its derivatives are considered strong antimicrobial and plant-strengthening agents in crop protection (reviewed [1]), comparable to biological control agents (BCAs), but their widespread application in industrial crop production is still pending. Due to the environmentally problematic production routes of chitosan and the poor understanding around cellular mechanism of action, there is a lack of broad applications in which potentially harmful substances, such as synthetic pesticides, could be substituted. A further benefit of using chitosan is its GRAS-status (generally recognized as safe), which would facilitate registration.

Systemic resistance induction of chitosan in plants has been studied over several years now (e.g., [3,4]). For instance, seed soaking, application on roots, or spraying of chitosan products elicit defense mechanisms in plants, thereby promoting pathogen resistance and concomitantly enhancing growth [1,2].

*Trichoderma* spp. are known to be promising tools in biocontrol. To this end, plant-beneficial *Trichoderma* species are primarily defined by their antagonism against plant pathogens, activation of the plant defense system [5] and as opportunistic plant symbionts that increase the systemic resistance of plants [6]. In fact, almost 90% of fungal BCA formulations and more than 60% of bio-based fungicides contain *Trichoderma* species [7,8] while *Trichoderma*-based organic fertilizers are becoming increasingly widespread due to their easier registration [9,10]. *Trichoderma* spp. are applied as seed, soil, or foliar treatments [5], bio-/mycofungicides, plant bio-stimulants [8], and organic fertilizers. Applying *Trichoderma* spp. to plants results in local and systemic defense reactions, which involves signaling cascades and activation as well as accumulation of defense-related antimicrobial compounds and enzymes in plants. The production of enzymes, such as pathogenesis-related (PR) proteins, phenyl ammonia lyase (PAL), and peroxidases, increased, and terpenoids, phytoalexins, and antioxidants (ascorbic acid, glutathione, etc.) are synthetized [9,11,12,13].

One of these promising BCAs is the mycoparasite *Trichoderma atroviride.* Mycoparasitic responses are triggered by molecules released from the host fungus (components released by cell wall degradation) or through physical contact, accomplished through surface-located components (e.g., lectins) [14]. *T. atroviride* inhibits or kills a host by hydrolyzing and parasitizing on its hyphae through cell wall degrading enzymes. Among these enzymes, the GH18 (endo-) chitinases (e.g., *ech42* during engagement of *Trichoderma* with *R. solani*, *chi 18-1*, and *chi18-13*), proteases (*prb1*, coding for a unique serine protease), the *N*-acetylglucosaminidase *nag1* [15], as well as glucanases are considered indicators for mycoparasitic activity. Recently, we have discovered that cell wall modifying enzymes, such as chitin deacetylases, chitosanases, and chitin synthases, are involved in cell wall remodeling during mycoparasitism and significantly contribute to the mycoparasitic capabilities [16]. This indicates that the first line of attack engaged by *T. atroviride* involves enzymes that can compromise the host’s cell wall integrity by targeting the glucan matrix but also its chitinous backbone. Fungal-secreted cerato-platanins are used to communicate with the plant and induce phytoalexin production and/or plant cell death in host and non-host plants, and are suggested as virulence factors in plant pathogenic fungi [17,18]. From this family of proteins, EPL1 was the first described non-enzymatic protein with an elicitor function in *T. atroviride* [19]. Moreover, secondary metabolites, including peptides and volatile compounds [20], act against fungal pathogens, by activating plant defense response to stress and/or improving plant root system morphology and physiology [7,10].

In recent years, a combination of BCAs with antimicrobial/biocidal compounds, such as chitosan, has become increasingly important [21,22]. In most of the studies, the combined effects of chitosan with other antimicrobial agents or BCAs are higher regarding the synergistic antimicrobial effects of chitosan. Particularly noteworthy for our study are the recently published results on the bio-efficacy of castor, groundnut, or safflower seeds, coated with a chitosan–polyethylene glycol *Trichoderma harzianum* (CHSN-PEG-*Th*) blend, which showed that the germination and vitality of the seedlings was significantly increased, but disease was reduced [21,23]. Moreover, a combination of chitosan (or a chitosan oligomer) and an albino strain of *T. harzianum* inhibited spore germination and, hence, the colony formation of the plant pathogens *Leptographium procerum* or *Sphaeropsis sapinea* [22].

Due to its high damaging potential, Cercospora leaf spot (CLS) disease caused by *Cercospora beticola* (class Ascomycota, ord. Dothideales, fam. Mycosphaerellaceae) is a continuous threat to sugar beet production worldwide. CLS disease severely affects sugar refinery in more than a third of sugar beet areas worldwide [24,25]. In Austria, sugar beets (*Beta vulgaris* ssp. *vulgaris*) are mainly cultivated in the eastern parts of lower Austria. The conditions for the main growth period of *B. vulgaris* in summer (June to August) are a warm, humid, continental, Pannonian climate with average temperature values of 18–20 °C, 80–90 mm monthly precipitation, and 10–11 h daily sunshine duration. These conditions and reduced or neglected crop rotation strategies promote the infestation with different fungal and insect-associated diseases.

The yield reduction due to CLS infestation is around 20 percent per year [25,26]. Usually CLS is combated by spraying the field with fungicides three to five times a year, negatively impacting the environment and the development of fungal resistance [25]. Immediate, natural, and safe application with effective and improved control of CLS disease is thus urgently needed. In addition, due to the climate crisis, new efforts should be made to support agriculture with new scientific findings on ecological and biocontrol strategies.

Despite its strong plant-beneficial efficacy, only a few studies dealt with application of *Trichoderma* strains to induced resistance against sugar beet pathogens so far [27,28,29,30,31,32] and only one investigated *Trichoderma* as a potential BCA against CLS [28]. In a recent study, the high potential of chitosan in combination with a commercially available resistance inducer (Acibenzolar-S-Methyl, ASM) as a foliar spray on table beet against CLS was demonstrated [33].

In the present study, the ability of *T. atroviride* IMI206040 and chitosan to promote growth and induce defense mechanisms in *B. vulgaris* against pathogens (*C. beticola*, CLS) was evaluated in planta and *in vitro*. We investigated seed coating and foliar application of the BCA *T. atroviride* and chitosan. Furthermore, host specificity of important sugar beet pathogens, such as *C. beticola*, *Fusarium* spp., *Penicillium* spp., and *Aspergillus versicolor*, was determined by employing chitosans of different molecular weight. Moreover, the combined application of *T. atroviride* and chitosan was assayed via a three-way interaction in plate assays *in vitro*. Furthermore, the defense stimulation against CLS disease in a greenhouse pot experiment was investigated.

Our findings expand the knowledge on *Trichoderma* spp. and chitosan as potent BCAs for the investigated pathogenic fungi in sugar beet production. We intend to evaluate novel combinations of BCA and chitosan derivatives with regard to their advantages and disadvantages on field application. To this end, CHSN and *T. atroviride* combined or in alternating application will become very attractive, if our understanding of chitosan, and its individual, potentiating, or synergistic effects, will be confirmed in biocontrol trials on field.

## 2. Materials and Methods

### 2.1. Strains and Cultivation Conditions

*Trichoderma atroviride* IMI206040 was maintained on potato dextrose agar (PDA, BD, Franklin Lakes, NJ, USA) and incubated at 28 °C with a 12 h/12 h light cycle.

*Fusarium culmorum*, *F. equiseti*, *F. oxysporum*, *F. sporotrichoides*, *Sclerotinia sclerotiorum*, *Botrytis cinerea*, *Rhizoctonia solani*, *Penicillium citrinum*, *P. crustosum*, and *Aspergillus versicolor* were maintained on PDA or malt extract (MEX, Roth, Karlsruhe, Germany) agar at 25 °C in the dark or stated otherwise. *S. sclerotiorum*, *B. cinerea*, and *R. solani* have been described previously [16,34]. The isolate *Cercospora beticola* 1437 is a verified isolate from a *B. vulgaris* CLS from a sugar beet field in lower Austria. The fungus was maintained on Czapek-Dox agar or PDA at 25 °C or otherwise stated. All other pathogenic strains were kindly provided by Agrana Research and Innovation Center (ARIC, Tulln, Austria).

### 2.2. Chemicals and Kits

Chemicals were obtained from Roth (Carl Roth GmbH + Co. KG, Karlsruhe, Germany) and Sigma-Aldrich (Sigma-Aldrich, St. Louis, MO, USA). A commercial preparation of chitosan of crustacean origin with different molecular weights was purchased from Sigma-Aldrich, including low-molecular-weight (LMW, CAS Number: 448869; DP: 50–190 kDa (based on viscosity), DA: 75–85%), high-molecular-weight (HMW, CAS Number: 419419; DP: 310–375 kDa, DA: >75%), and practical-grade (PG, CAS Number: 417963; DP: 190–375 kDa, DA: ≥75%) chitosan. Enzymes and kits were obtained from Thermo Fisher (Thermo Fisher Scientific, Waltham, MA, USA), Promega (Promega Corporation, Fitchburg, WI, USA), QIAGEN (QIAGEN N. V., Venlo, The Netherlands), and Bio-Rad (Bio-Rad Laboratories, Hercules, CA, USA).

### 2.3. Cultivation Conditions for T. atroviride and Plant Pathogenic Fungi on Plates and in Liquid Medium

Chitosan of different grades (LMW, HMW, and PG) were prepared as 20 mg mL^−1^ stock solutions in 100 mL of distilled water (pH 7.0) and the pH was adjusted with dilute acetic acid to 5.6 [35]. Working concentrations were chosen based on previous experiments, with 2 mg mL^−1^ of HMW chitosan appearing to be the highest soluble concentration in plate assays. After stirring the chitosan stock solutions over night at RT, they were used to prepare PDA plates with a final concentration of 2 mg mL^−1^ of chitosan. As a control condition, PDA was supplemented with the same amount acidified water, pH 5.6. A 5 mm disc containing the pathogen mycelium or *T. atroviride* was placed in the center of each petri dish. The plates were incubated at 28 °C in a light/dark cycle. The radius of the colonies of the pathogens was measured under all conditions before the fungi reached the plate edge on PDA (control condition) or when sporulation was initiated. The radius of *T. atroviride* was recorded after 3 and 6 days. The experiment was carried out with two biological replicates and two technical replicates, and representative pictures are shown.

In order to determine the germination efficiency, *C. beticola* conidia were harvested from 10-day-old plates, and ca. 2.5 × 10^2^ conidia were plated on PDA containing 2 or 1 mg mL^−1^ of HMW chitosan and compared to the control condition. The plates were incubated for 3 and 6 days at 25 °C in the dark and colony-forming units from single spores (CFU mL^−1^) were determined.

For RNA extraction from *T. atroviride*, the fungus was either grown on different chitosans or *N*-acetyl glucosamine (Sigma-Aldrich). Media supplemented with glucose or glycerol were used as controls (Sigma-Aldrich). A liquid SM culture medium was used for liquid standing cultivation, as described previously [16], and supplemented with 10 mg mL^−1^ of the indicated polymers or monomers. Then, 1 × 10^6^ *T. atroviride* spores were used to inoculate the medium. For growth on cellophane discs, PDA plates were covered with a sterile cellophane disc prior to inoculation with *T. atroviride*. Mycelium was harvested after 24 h of growth at 28 °C by filtering through miracloth (Millipore/Sigma, VWR), followed by excessive washing, and frozen immediately in liquid nitrogen.

### 2.4. Confrontation Assays

Dual confrontation assays of *T. atroviride* against *B. cinerea, S. sclerotiorum*, *F. oxysporum*, or *C. beticola* were performed as described in [34]. To evaluate synergistic effects between *T. atroviride* and chitosan, PDA medium was supplemented with chitosan solutions of different molecular weights (HMW, LMW, PG, 2 mg mL^−1^). As controls, PDA plates with an equal volume of acidified water, adjusted to pH 5.6, were used. Agar plugs (5 mm) were taken from *T. atroviride* and the hosts were placed on a 94 mm petri dish approximately 50 mm apart. The growth of the hosts *B. cinerea, S. sclerotiorum*, and *F. oxysporum* was monitored after six days in circadian illumination at 28 °C. Due to its slow growth, *C. beticola* was inoculated six days prior to *T. atroviride* and measured on the twelfth day of incubation. The experiment was carried out with two biological replicates, and two technical replicates and representative pictures are shown.

### 2.5. In-Vitro Cultivation of B. vulgaris Seedlings and Germination Efficiency

For *in vitro* cultivation of *B. vulgaris* seedlings (cultivar: Hannibal DE068-001723319) 100 native seeds were coated for 2 h at RT with constant shaking (800 rpm), either with 10 mg mL^−1^ of HMW chitosan or distilled water (pH 5.6). The treated seeds were dried overnight on filter paper.

Adapted to the rules for seed testing of the International Seed Testing Association [36], 100 seeds per condition were distributed in 5 Petri dishes (20 seeds per Petri dish) and placed on a layer of moistened filter paper. The seeds were incubated in the dark at 22 °C. The germination percentage (GP), indicating the percentage of germinated seeds after 48 h, 72 h, and 120 h (1); the germination capacity (GE, in percent), indicating the germination efficiency per day (2); and the germination index (GI, in percent), as a mean of the total germination capacity over time (3), were examined [36]. For calculation purposes, the below formulas were used, where A is the total number of seeds, and B, C, and D are the number of seeds germinated after 48 h, 72 h, 120 h, respectively. F is the number of days of seed germination.
GP = B, C or D/A × 100(1)
GE = B, C or D/A/F × 100(2)
GI = Average (B/A + C/A + D/A) × 100(3)

Sterile conditions were used to cultivate seedlings for RNA extraction. For this purpose, the seeds were surface-sterilized with a 70% alcohol solution for 5 min, followed by further sterilization without prior rinsing with a sodium hypochlorite solution (0.2% BayroChlorit) diluted in distilled water and 4 drops of Tween-20 (in 100 mL). After 20 min of manual shaking, the seeds were rinsed three times with sterile, distilled water and coated either with 10 mg mL^−1^ of chitosan, a 10^4^ *T. atroviride* spores mL^−1^ solution containing 10 µL mL^−1^ of glycerol, or distilled water pH 5.6. Five coated seeds were placed 25 mm apart in the upper quarter of a square Petri dish with a length of 120 mm filled with 0.6% phytagel, supplemented with 10 g L^−1^ sucrose, 0.4 g L^−1^ CaCl_2_·2H_2_O. The seeds were preincubated at 4 °C in complete darkness for 48 h.

The cultures were grown at 24 °C with a 12 h/12 h circadian rhythm. Seedling material was harvested from roots and shoots of every condition after 10 days of incubation, and material was immediately frozen in liquid nitrogen and stored at −80 °C for RNA extraction. The experiment was repeated twice.

### 2.6. Greenhouse Pots Experiment of B. vulgaris and C. beticola Infection

The plants were grown on a conventional peat-based pot mix in a small greenhouse test setup over a period of 3 months from May to August 2021 without additional heating or light. The average perennial (1981-2010) monthly temperature and hours of sunshine for the region around Wiener Neustadt (local experiment) are as follows: May—15.2 °C and 229 h; June—18 °C and 228 h; July—20.2 °C and 250 h; and August—19.8 °C and 239 h. During the test period, the monthly average temperature was −1.5 °C lower in May and +3.6 °C lower in June and July, compared to the annual average. In the greenhouse experiment, an irrigation scheme with 100 mL of tap water was carried out every second day. To investigate the growth and/or defense stimulating effects of each treatment, the effects of all agents and control condition were preventively sprayed on whole plants.

Seeds were germinated in small containers and transplanted after 2 weeks (2-cotyledon stage) to larger pots (one plant per pot), with 180 mm diameter. Seedlings were then cultivated for 3.5 more weeks to the six-leaf stage. The treatments were applied to six expanded beet seedlings obtained from cultures selected on the basis of their synchronized cotyledon developmental stage. At the age of 37 (t1), 57 (t2) and 70 (t3) days, plants (*n* = 6) were treated with a hand sprayer (volume of 1 l). The plants were sprayed with 10 mL per plant from the following preparations: 1 mg mL^−1^ of HMW chitosan (CHSN, Sigma-Aldrich; pH 4.5), freshly harvested *T. atroviride* spores from a 7-day-old culture on PDA (TA.SP, 4 × 10^4^ spores mL^−1^), or distilled H_2_O as control (acidified with acetic acid, pH 4.5). Twenty and thirteen days after treatment (t1 and t2, respectively), one leaf was collected from each plant and leave tissue was harvested for RNA extraction and expression analysis. Consequently, photographs were taken for visual assessment of vigor and strength. Growth promotion, vitality, and frass were documented.

Seven days after the third treatment, all plants were inoculated with a freshly prepared spore solution of *C. beticola*. The spores were obtained by adding 10 mL of sterile distilled water to a petri dish containing a 10-day-old culture of *C. beticola*, which was then scraped off with a Drigalski spatula, and the spore concentration was adjusted to 10^4^ spores mL^−1^. After spraying each plant with 10 mL of spore solution, the plants were covered with transparent polyethylene bags (50 cm × 100 cm) to maintain humidity. The severity of the disease was determined 2 and 3 weeks after the inoculation. The disease severity and degree of invasive growth of CLS were documented by means of photography and visual assessment of the developed leaf spots and frass. Disease incidence was assessed as described:Disease incidence (%) = (number of infected leaf/plant/total number of leaf per plant) × 100.
The incidence of frass (%) = (total number of leaves showing signs of frass or skeletal frass/number of leaves of all plants) × 100.

### 2.7. Leaf Disc Assay for the In Vitro Assessment of CLS Disease

The *in vitro* leaf disc assay was performed essentially as described. Adult plants at the 8-leaf stage were treated with 1 mg mL^−1^ of HMW chitosan 24 h prior to the assay start. Leaf discs with a 10 mm diameter were cut from the leaves, placed with the upper surface facing upwards on water agar (0.5%), and sprayed with a 10^4^ spores mL^−1^ solution of freshly harvested *C. beticola* spores. The plates were sealed with parafilm and incubated at 22 °C in a light/dark cycle for 14 days. The disease severity on each leaf disc was evaluated by counting the number of spots per leaf disc according to a scale with five categories: A = discs with no CLS disease; B = discs with little disease (up to 5 spots); C = 6–10 CLS per disc; D = 11–20 CLS per disc; and E = leaf disc covered entirely with spots. The number of discs for each category was assessed for CHSN-treated and control leaves.

### 2.8. RNA-Extraction and cDNA Amplification

For RNA extraction from *T. atroviride* ca. 100 mg of mycelium were ground to a fine powder with glass beads (2 beads with a diameter of 2.85–3.45 mm and 200 µL of beads with a diameter of 0.75–1.0 mm) and a bead mill (2 × 60 s at 6 m/s; Bead Ruptor 24 Elite Bead Mill Homogenizer, Omni International, VWR). For samples from *T. atroviride* grown on chitosans, the total RNA was extracted using an extraction protocol adapted by [37] suited for RNA recovery from samples grown on agar to reduce binding of chitosan to the RNA. For RNA extraction from *T. atroviride*, samples grown on other carbon sources (glucose, GlcNAc, glycerol), on cellophane discs, or isolated from *B. vulgaris* seedling roots, the QIAGEN RNeasy Mini Kit for plant and fungal tissue, including the QIAshredder^®^ cell-lysate Homogenizers (QIAGEN N.V.), was used. For RNA extraction from plant material, either ca. 100 mg of frozen *B. vulgaris* roots or shoots from seedlings (usually 3-4 seedlings) or a leaf area corresponding to ca. 100 mg were ground to a fine powder in a bead mill (1 × 60 s at 3200 m/s) using 2 stainless steel beads (diameter 3 mm). The QIAGEN RNeasy Mini Kit was used to extract RNA. RNA was treated with DNAse I and purified with RNA GENEJet Micro-Cleanup kit (Thermo Fisher Scientific, Waltham, MA, USA), and the RNA quality was evaluated for purity and integrity with the TECAN NanoQuant Plate™ (Tecan Group Ltd. Männedorf, Switzerland). cDNAs were generated from 1 µg of RNA with the Revert Aid H-minus cDNA synthesis kit (Thermo Fisher Scientific, Waltham, MA, USA).

### 2.9. Gene Expression Analysis

For gene expression analysis of *T. atroviride* cultured on chitosan of different molecular weight and control substrates, or isolated from *B. vulgaris* seedling roots or cellophane, RT-PCR reactions were performed, using GoTaq^®^ DNA Polymerase from Promega (Promega Corporation) and a T100 Thermal Cycler (Bio-Rad), as described previously [38]. The primer pairs are described in Appendix A. PCR products were analyzed by agarose gel electrophoresis using a Gel Doc™ system (Bio-Rad).

For gene expression analysis of *B. vulgaris* seedlings and young plants, RT-qPCR reactions were performed in a Rotor-Gene 6000 (QUIAGEN N.V.) as described previously [39]. Relative gene transcript levels were quantified and normalized to the corresponding signals of the *B. vulgaris* housekeeping genes ACT7, TUB1, and GAPDH. All primers are listed in Appendix A. The fold change relative to the control conditions was calculated using the ΔΔCT method with REST software [40].

### 2.10. Microscopic Analysis

Strains grown on PDA or PDA with 0.6 mg mL^−1^ of LMW chitosan were applied in a droplet of 30 μL of water with the inverted agar method and imaged with an inverted Zeiss Axio Observer Z1 (Zeiss, Oberkochen, Germany) with differential interference contrast optics.

### 2.11. Statistical Analysis

Statistical analysis for RT-qPCR was performed with REST software with a pair-wise fixed reallocation randomization test. At least six biological and three technical replicates were used for statistical analysis of RT-qPCR data. For evaluation of the germination ability of *C. beticola*, statistical analysis was performed using Poisson distribution. For all other statistical analyses the Student’s t-test was used, assuming unequal variance of groups.

## 3. Results

### 3.1. Chitosan Strongly Impairs the Growth of Important Plant Pathogens

We evaluated chitosan with different molecular weights from commercial preparations (see Materials and Methods) on growth inhibition of important phytopathogens in crop production, with special emphasis on *B. vulgaris* cultivation. The investigated fungal pathogens included four *Fusarium* spp., two *Penicillium* spp., *A. versicolor*, *S. sclerotiorum*, *B. cinerea*, *R. solani,* and *C. beticola*, which are known threats in agriculture (food spoilage, horticulture, post-harvest processing).

The strains were cultured on solid PDA plates with 2 mg mL^−1^ of chitosan of various grades. Representative images of all conditions are shown, under which the pathogens under the control condition (PDA, control) reached the edge of the Petri dish or started to sporulate (Figure 1).

Our results confirmed the strong biocidal effect of various chitosan grades and showed a high fungicidal effect on almost all selected pathogens. LMW and HMW chitosan were most effective and showed 60 to 80% growth inhibition for *C. beticola* and *B. cinerea* and between 80% and nearly 100% inhibition for the investigated *Fusarium* spp. over a prolonged incubation time of up to 12 days. *Penicillium* spp. growth was also strongly inhibited by LMW and HMW chitosans. Practical-grade chitosan (PG), on the other hand, was less effective to inhibit growth of these species.

*R. solani* was only moderately inhibited (20%) by all grades of chitosans and *S. sclerotiorum* showed 35% inhibition on HMW chitosan and around 50% inhibition by LMW and PG chitosans. Despite the fine distribution of the crystal spores on the plate of *Penicillium* spp. and *A. versicolor* (and other investigated *Aspergillus* spp., data not shown), the latter were not or less affected by chitosan.

Next, the strong growth inhibition of *F. oxysporum* and *C. beticola* on the fungal morphology was confirmed in microscopic examinations on solid PDA. Chitosan led to hyphal agglomeration in *F. oxysporum*. *C. beticola* showed a dense hyphal network and a short, narrowed growth phenotype, which indicates a strong growth inhibition by 0.6 mg mL^−1^ of chitosan (Appendix A). Compared to the control condition, hyphal growth was severely impaired, with abnormal shape, dense network formation, and a short, nodular, concentric growth phenotype. This confirms the strong growth inhibition of chitosan in both pathogens, also on a microscopic level.

### 3.2. Combined Treatments of T. atroviride and Chitosans Reveal Additive Fungicidal Effects on Selected Plant Pathogens

In order to assess if a combination of *T. atroviride* and chitosan could potentiate the inhibition of phytopathogens, we first needed to assess if *T. atroviride* is able to grow on medium supplemented with the chitosans. *T. atroviride* was placed on PDA containing 2 mg mL^−1^ of the chitosans and growth was monitored after 3 and 6 days. Interestingly, *T. atroviride* growth was initially impaired, but the whole plate was covered with mycelium after 6 days, followed by the production of mature green conidia (Figure 2A,B). We conclude that *T. atroviride* can establish itself in the presence of high concentrations of chitosan with different grades. By contrast, in the plant pathogens, which were sensitive to chitosan, we did not observe an adaption to chitosan even after prolonged incubation times.

Next, we compared the colonial growth of the pathogens *C. beticola*, *F. oxysporum*, *B. cinerea,* and *S. sclerotiorum* in the presence and absence of *T. atroviride* (PDA control) to PDA medium supplemented with one of the three chitosan grades before contact and at contact. The results showed a strong increase in the fungicidal effect by a combination of *T. atroviride* and HMW or PG chitosans. Co-cultivation of *T. atroviride* on control medium PDA and medium supplemented with PG chitosan resulted in a similar increased inhibition of all tested pathogens. Most importantly, the best effectiveness against *C. beticola* was observed with HMW chitosan, with a further increase in inhibition by the presence of *T. atroviride* by 17%, leading to an overall inhibition of 93% (Figure 2C,D). HMW chitosan and *T. atroviride* induced a stable fungistasis on *C. beticola*. The combination of HMW chitosan with BCA also led to a 19% increased growth inhibition in *S. sclerotiorum*, but only a 50% total inhibition, similar to the inhibition by the other tested chitosans (LMW or PG). The growth inhibition of *B. cinerea* and *F. oxysproum* by HMW chitosan could not be strongly increased in dual confrontation with *T. atroviride*. In addition, the combination with LMW chitosan did only show a small or no additive or synergistic effect in the tested strains. It is important to note that the inhibition by LMW and HMW chitosan alone is already very high for these pathogens (Appendix A and Figure 2D), and a further increase is thus most probably not visible with the high applied CHSN concentration. On the other hand, an increase in growth inhibition with LMW or HMW chitosans was less pronounced in *B. cinerea*, but a strong effect of *T. atroviride* and additive chitosans on the growth of *B. cinerea* and *S. sclerotiorum* is also shown by the fact that the pathogens seem to try to escape the antagonist, which can be observed at the site of confrontation of the *B. cinerea* colony on LMW, compared to the control condition (arrows, Figure 2C).

From these findings, we conclude that the presence of chitosan in the medium could additionally trigger the fungus to secrete hydrolytic enzymes, which renders it more aggressive towards phytopathogens.

To prove this hypothesis, we investigated the enzymatic transcription profile of *T. atroviride* grown on medium supplemented with the different chitosan grades.

The expression of the chitosanases *cho1*, *cho2*, *cho3*, and *cho5*, but not of the chitin deacetylases (CDAs) with the exception of the conidiation-associated *cda3* [16], was strongly induced in the presence of chitosans (Appendix A). Furthermore, mycoparasitism markers *chi18-5* and *nag1* were strongly upregulated. Interestingly, these genes were also strongly upregulated after contact with host pathogens [14,16]. On the other hand, the mycoparasitism genes *cda1* and *cda5* were not stimulated. Our results show that a combined application of CHSN and *T. atroviride* leads to an increased expression of secreted enzymes in the mycoparasite and increased inhibition of growth of the tested plant pathogens.

### 3.3. Chitosan Promotes Germination of B. vulgaris

To assess the effect of chitosan on seedling development, seeds were coated with 10 mg mL^−1^ of HMW chitosan or acidified water (pH 5.6) as control. We assessed germination capacity (GE) and the germination percentage (GP) after 48 h, 72 h, and 120 h, and calculated the germination index (GI) thereof (Figure 3A, Table 1). The germination percentage reflects the viability of a population of seeds and was significantly higher on the 2nd, 3rd, and 5th day in chitosan-treated seeds compared to the control conditions with 15% +/− 5 increased germination after 5 days (Figure 3A). Germinability exceeded that of the control condition. An “earlier” or accelerated onset of germination (33% faster after 48 h) by chitosan priming, in comparison to control conditions, was observed (Table 1). In addition, the germination index was 17.38% from chitosan-coated seeds compared to 13.02% in the control. Moreover, we assessed the fresh weight of the seedlings and found a 17% increase in fresh weight of the seedlings of chitosan-coated seeds compared to the control condition (Figure 3B). Interestingly, this increase in fresh weight appeared to be due to increased root elongation as opposed to shoot development, since the roots and hypocotyl showed 30% higher weight than the control conditions, while the shoots did not differ from the control (Figure 3C). In our experiments, chitosan coating increased germination and promoted the early onset of seedling development under semi-sterile conditions.

### 3.4. Foliar Application of Chitosan or T. atroviride Spores on B. vulgaris

#### 3.4.1. *T. atroviride* and Chitosan Positively Affect Growth-Promotion of *B. vulgaris*

In order to compare the effect of both bioactive compounds, young *B. vulgaris* plants in the six-leaf stage were treated with three different preparations: 1 mg mL^−1^ of HMW chitosan (CHSN, pH up to 4.5); *T. atroviride* IMI206040 spores (TA.SP, 10^4^ spores mL^−1^); and, as a control, distilled water, adjusted to pH 4.5 (control). Plant growth and vitality were monitored and documented (Figure 4A–C) with three treatments over 40 days. Both the leaf treatment with the *T. atroviride* spore solution and with HMW chitosan resulted in faster and stronger growth, a more intense leaf color, and a larger leaf area, compared to the control conditions (Figure 4A,B). This effect was amplified with the second (Figure 4C) and third stimulus (data not shown).

After the treatment with CHSN, strong, green, and “glowing” plants, in particular, were observed. We assume that the vigorous growth could be caused by a stable high turgor in the treated plants.

#### 3.4.2. Chitosan and *T. atroviride* Activate Differential Expression Patterns of Defense-Related and Plant Growth Promoting Genes in Sugar Beet Leaf Tissues

Next, we were interested in the expression profile of treated *B. vulgaris* plants. From the greenhouse pot experiment (Section 3.4.1), leaf tissue was collected 20 and 13 days after the first and second treatment, respectively, and RNA expression profiles for selected marker genes were analyzed by quantitative real-time PCR (RT-qPCR). The analysis revealed that only a small set of the probed indicator genes is differentially regulated by the foliar treatments and that the expression profiles of the two agents do not completely overlap. First, we observed a stable activation of the basic class IV chitinase, a gene coding for a pathogenesis-related PR-3 enzyme [32], by both the BCA and the biocidal HMW chitosan, 13 and 20 days after the two consecutive treatments (Figure 4D, Appendix A).

These data indicate that systemic resistance is already induced in *B. vulgaris* after the first treatment of foliar spray with either chitosan or *T. atroviride* and stably maintained after the second treatment, indicated by the stable upregulation of the PR-3 indicator gene.

In contrast to application of the *T. atroviride* spores, priming the plant with HMW chitosan led to a significant upregulation of two more groups of defense-related genes. The oxidative burst indicators glutathione-S-transferase (GST) and 6,7-dimethyl-8-ribityl lumazine synthase (DMRL) [41,42] were upregulated 20 days after the first treatment with chitosan, and GST showed a stable induction also after the second treatment (Figure 4D).

Furthermore, especially after the first treatment with chitosan, the expression levels of phenylalanine ammonia lyase (PAL) (EC 4.3.1.5) [43] and aspartate aminotransferase (AAT) [41] were increased (Figure 4D).

PAL and GST stayed stably upregulated in response to treatment with chitosan application on leaves; even after two and three weeks, but no response toward *T. atroviride*, treatment was observed after leaf application, indicating that the polymer might be able to more efficiently prime the plant for a potential infection by a plant pathogen than the spores.

Our findings show that chitosan similar to *T. atroviride* spore application on leaves activates a distinct pathogenesis-related cascade (PR-3), but only chitosan induces the expression of two more groups of genes related to oxidative stress, defense, and plant-growth-promoting pathways.

#### 3.4.3. Priming with CHSN and *T. atroviride* Induces Expression of Defense Indicator Genes in *B. vulgaris* Seedlings

We were also interested whether chitosan coating also triggers the expression of indicator genes for plant health and resistance to pathogens in seedlings. In a second approach, we supplemented the medium with chitosan, instead of seed coating. In a third approach, seed coating with 10^4^ *T. atroviride* spores mL^−1^ was compared (Materials and Methods). Although the beneficial effects on germination and growth by chitosan were similar to the filter paper assay (data not shown), the expression of the selected set of genes was not significantly altered in the seedlings from chitosan-coated seeds in comparison to uncoated seeds (Figure 5A). Thus, in contrast to foliar application, seed coating with CHSN alone does not induce systemic resistance of the examined defense genes after 10 days of germination.

Interestingly, although the reaction of both parts of the seedling was comparable in seedlings from CHSN-coated seeds, we noticed a strong difference in the expression pattern of roots and shoots when seedlings were instead cultured in medium containing chitosan (Figure 5B). In the roots and hypocotyl of the seedlings, all tested genes were strongly upregulated, including those related only to short-term induction (GLU2 and SE2), indicating that the polymer serves as a signal for the presence of a pathogen and concurrently primes the seedling for a potential attack. In the shoots, except for PR-3, which was highly upregulated, the investigated gene expression was unaltered or slightly downregulated, suggesting induced systemic resistance (PR-3) in the above-ground parts of the seedling, transmitted from signals originating in the roots. A stronger root hair development was observed in many of the collected seedlings in comparison to the control seedlings (data not shown), even though the roots were shorter than the ones from the control and the chitosan-coated seeds. The stronger root hair formation induced by the presence of chitosan might, in turn, help the plant to mobilize nutrients from the medium more efficiently and, thus, might promote the growth of the seedlings.

Interestingly, when grown from *T. atroviride* spore-coated seeds, 10-day-old seedlings showed strong upregulation of all genes, with the exception of PAL in both roots and shoots. To assess if *T. atroviride* established communication between the BCA and the plant, we extracted *T. atroviride* mRNA from the root tissue of *B. vulgaris* and compared it to the growth on cellophane as surface. Gene expression analysis revealed that *epl1*, an elicitor of local and systemic disease resistance [19], was highly expressed in *T. atroviride* isolated from the roots of *B. vulgaris* (Appendix A).

#### 3.4.4. CHSN Primes for Resistance to *C. beticola* in a Greenhouse Pot Experiment

To assess if foliar application with either chitosan or *T. atroviride* spores primed *B. vulgaris* for resistance against fungal pathogens, we challenged the primed and control *B. vulgaris* plants with *C. beticola* spores.

Compared to the control group, plants treated with CHSN were 47% less infected and the *T. atroviride* spore group was 27.9% less infected (Table 2). Striking differences were also found in the type of CLS infestation. The control group showed strong and later completely necrotic leaf tissue, while the *T. atroviride*-treated group had small, concrete, dark spots with generally less necrotic tissue. The group treated with CHSN showed less invasive and light-brown-colored CLS with little necrotic tissue, indicating the delayed establishment of the phytopathogen (Figure 6A). After repeated foliar spray over a period of ca. 40 days, particularly with chitosan, CLS was less drastic.

The results clearly indicate that foliar application with HMW chitosan and *T. atroviride* spores effectively decreased the severity of CLS in *B. vulgaris*. Remarkably, the strong CLS infestation observed in the control group also appeared to influence the general susceptibility of the plants to insect frass. Several leaves, in addition to the coalescent and partially collapsed leaf areas originating from advanced CLS, also showed signs of herbivorous insect pests, i.e., leaf miner damage. By contrast, in the CHSN and *T. atroviride* spore-treated plants, we observed almost no damage by insect pests.

To quantify the disease resistance in CHSN-treated plants in an independent setup, control plants and plants with foliar spray of chitosan were used in an *in vitro* assay. *B. vulgaris* plants were either sprayed with HMW chitosan or acidified water before infection with *C. beticola*. After 24 h from the primed leaves, round discs (10 mm diameter) were cut from the whole leaf area and treated with a *C. beticola* spore solution. After 14 days, the emergence of disease symptoms was evaluated. The symptomatic leaf areas were categorized into five categories from 0 CLS per disc to more than 20 CLS per leaf disc, and the number of discs for each category from treated and control leaves was compared. A strong reduction in disease severity was observed in chitosan-treated leaves in comparison to leaves treated only with water (Figure 6B). Whereas all leaf discs from the control condition showed CLS, and 20 out of 23 leaf discs showed severe CLS above six spots per disc, only 14 out of 22 leaf discs from chitosan-treated plants fell into these higher categories and 2 discs did even not show any signs of disease (Figure 6B).

In order to confirm the fungistatic effect on *C. beticola* in the performed foliar application, *in vitro* cultures for spore germination were carried out. Germination was markedly reduced in the case of PDA plates containing 1 mg mL^−1^ of HMW chitosan; only 33.57% ± 3.72 colonies could be formed from spores of *C. beticola* compared to growth on PDA control plates (Figure 6C, representative pictures shown in Appendix A). At a concentration of 2 mg mL^−1^, the germination of *C. beticola* was completely inhibited.

## 4. Discussion

*C. beticola* causes one of the most important and devastating leaf diseases in the plant industry worldwide [24,25], whereby CLS disease is a particular burden for sugar farmers and thus also for exploitation in the processing industry in Austria and Europe. The frequent application of fungicides, mainly of chemical-synthetic origin, continues to be the primary means of combating CLS disease. An effective way to combat disease is to generate hybrid species with increased genetic resistance, though they might have lowered yields or decreased final product quality. Not least in order to conform to the rules of the Sustainable Development Goals (SDG 2030), efforts have been made in recent decades to promote sustainable BCA.

In this study, we examined and compared two of these promising alternatives to common fungicides, namely the biocidal natural product chitosan and the BCA *T. atroviride* IMI206040. Our results confirmed their high potential in plant growth promotion of *B. vulgaris* and provided new insights into the strong reduction in CLS disease. Our results demonstrate that LMW and HMW chitosan at a concentration of 2 mg mL^−1^ in solid-plate assays are highly effective against a variety of important fungal crop pathogens, including four *Fusarium* spp., two *Penicillium* spp., *B. cinerea*, and *C. beticola*. This is in line with other recently published findings on fungal pathogens, including *F. oxysporum* [44] and other *Fusarium* spp., *B. cinerea*, *Rhizopus* spp., *Phytophthora* spp., and *Alternaria* spp. [45]. *R. solani* showed only a 20% inhibition by all grades of chitosans, which confirmed the findings described in [46]. Only *Aspergillus* spp. were not strongly affected by any of the chitosans in our *in vitro* assay, although an inhibition by COS has been described [47]. The comparison of the three different chitosan grades enabled us to select the most effective among all the pathogens examined. PG chitosan showed the least efficacy, but LMW and especially HMW chitosan turned out to be the most promising biocidal agents with high fungicidal effects on almost all selected pathogens important for agriculture (food spoilage, horticulture, post-harvest processing). Inhibition of between 80% and nearly 100% for *C. beticola* and all *Fusarium* spp., respectively, was observed. Previous studies showed improved antifungal efficacy of chitosan oligomers on some fungi that increase with polymer size [48]. The strong biocidal effect of chitosan leads to hyphal agglomeration in *F. oxysporum* solid cultures [45], which we also confirmed microscopically in this work. In general, negative autotropism allows the fungus to explore and adapt to the existing substrate, while subapical parts of the hyphae form new branches. Chitosan could form an impermeable polymer layer that changes cell permeability and prevents nutrients from entering the cell. This can lead to a loss of hyphal avoidance and, thus, a reduced perception and utilization of the available substrate, which leads to loss of the negative autotropism. Due to the lack of nutrient absorption, which is prevented by chitosan (see hyphal bundles, Appendix A), hyphae stop growing and, therefore, show abnormal shapes. *C. beticola* shows reduced germination on plates containing chitosan with hyphae showing a short, narrowed growth phenotype that indicates a strong growth inhibition by chitosan. This results in a dense network of hyphae with a short, nodular, and concentric phenotype. Thus, microscopic data confirmed the strong growth inhibition of chitosan for both severe pathogens.

One important trend in biocontrol is the application of combinations of BCAs or mixing a BCA with a biocidal natural agent [21,23]. Evaluating their potential synergism, additive effects, or antagonism before application is important since a shift towards increased, reduced, or similar effectiveness determines their best way of application and combination. To this end, we started to investigate how chitosan affects the mycoparasitic behavior of *Trichoderma in vitro.*

Interestingly, in a confrontation assay for a combined activity of chitosan and *T. atroviride*, we observed a strong synergistic effect for the growth inhibition of momentous plant pathogens, including *C. beticola* and *F. oxysporum.* The combination of HMW chitosan with *T. atroviride* induced the most stable fungistasis in *C. beticola*, increasing by 17% to an almost total inhibition of 93%. To the best of our knowledge, our study is the first report on the use of *T. atroviride* as an efficient mycoparasite for *C. beticola* biocontrol. It is also important that the otherwise weaker inhibition by PG chitosan could be significantly increased in the combined confrontation assay for all investigated pathogens. The weaker inhibition is most likely due to its less defined nature, and the increase in inhibition is thus mainly due to the presence of *T. atroviride* and only a small additive effect of the PG chitosan itself. We assume that an additive effect caused by chitosan triggers an increased production of secreted hydrolytic enzymes in *T. atroviride* and, thus, other potential effectors, which renders it more aggressive towards phytopathogens. Indeed, the mRNA levels for four secreted chitosanases and a chitin deacetylase, which we also recently identified as highly upregulated after contact with host pathogens [16], were strongly elevated. Furthermore, important mycoparasitism marker genes, such as *ech42* and *nag1*, were upregulated. Based on these results, we believe that the chitosan polymers mimic the presence of a potential host for the antagonist *T. atroviride*, thus triggering the production of a subset of secreted enzymes that enable parasitism and killing of the host, which might also explain why *T. atroviride* can adapt to growth on chitosan. Chitosan inhibits the host due to a direct biocidal mechanism, while growth inhibition by the presence of *T. atroviride* is most likely due to secretion of secondary metabolites and enzymes after sensing of the host [16]. To this end, chitosan might increase the susceptibility of the hosts to the indirect antagonistic attack by a BCA.

Since especially HMW chitosan proved highly effective against the majority of investigated plant pathogens, we were also interested in the impact of HMW chitosan on the sugar beet cultivar *B. vulgaris*. HMW CHSN, an elicitor of defense mechanisms in plants, has been shown to promote pathogen resistance and growth (reviewed in [1]). *T. atroviride* IMI206040 can synthesize IAA [49] or volatile organic compounds (mVOCs), e.g., 6-pentyl-2H-pyran-2-one (6-PP), which are described in promoting plant growth [50].

We designed a greenhouse pot experiment with three treatments of foliar spray over a period of 40 days with chitosan or *T. atroviride* IMI206040 spores on young *B. vulgaris* plants in the six-leaf stage. The *T. atroviride* spore-treated plants, and especially the chitosan-treated plants, showed significantly promoted growth and vigor, with intensified leaf color and larger leaf area. Importantly, every treatment increased the observed positive effects on plant health. *Trichoderma* application increases the tolerance to abiotic stress (reviewed in [7,13]), such as drought [11] and salinity stress [51]. The plants in the greenhouse experiment faced high temperatures during July and August, and chitosan treatment has been shown to improve tolerance to heat stress as well [52].

In plants treated with chitosan, the elicitation of phytoalexin and phenolic precursors, an increased production of chitinases, and reduced levels of aflatoxin together with other factors relevant for plant defense have been described [1,2]. Notably, a successful test for preventive chitosan treatment of table beet to decrease CLS was carried out recently [33]. Moreover, foliar spray with *Trichoderma* spp. isolated from sugar beet fields reduced the disease incidence by *C. beticola* in sugar beet [28]. Strikingly, in our greenhouse experiment, the repeated foliar application with either *T. atroviride* spores or chitosan elicited defense responses in *B. vulgaris* and considerably reduced CLS. The plants primed with chitosan were 47% less infected and the *T. atroviride* spore group was 27.9% less infected by CLS, and the CLS tissue was less necrotic than in the control group. Apart from eliciting defense responses in plants, chitosan itself exerts high, intrinsic, antifungal activity (reviewed e.g., [1]), which was also confirmed with the leaf disc assay and the *in vitro* germination assays for *C. beticola*. We propose that a simple application of repeated foliar spray with the potent antifungal and defense-inducing agent chitosan is sufficient to considerably reduce CLS in sugar beet.

We would also like to draw the attention to an additional observation, that the strongly compromised control group showed signs of a generally increased susceptibility to frass by herbivorous insect pests, i.e., leaf miner damage. Almost no damage by insect pests was seen in the chitosan or *T. atroviride* spore-treated groups. A similar effect was reported in chitosan coated soybeans against *Agrotis ipsilon*, soybean capsule borer, and soybean aphid [36].

Our first trials with both active ingredients had a positive effect on the development, growth promotion, and induction of systemic resistance in *B. vulgaris*. We further show that, instead of inoculating the soil with *T. atroviride* spores, foliar spray triggers the defense mechanism of *B. vulgaris* and also strongly increases resistance towards invasion by the most notorious pest *C. beticola*. We confirmed that foliar application of young plants with either *T. atroviride* spores or 1 mg mL^−1^ of HMW chitosan, resulted in priming for a future attack, indicated by a stable induction of the expression of a pathogenesis-related gene (PR-3), a basic class IV chitinase, weeks after application. A systemic induction of chitinase activity in sugar beet leaves after treatment with different *Trichoderma* isolates was connected to a reduction in CLS disease, as noted previously [28].

By contrast, the acidic class III chitinase (SE2) and a basic GH17 beta-1,3-glucanase (GLU2) were downregulated with both agents. Nielsen 1994 [53] showed that, in sugar beet, the induction of some PR proteins, including the chitinase SE2 and glucanase GLU2, is in contrast to other plants not correlated with the induction of resistance. Furthermore, GLU2 has been identified as a strong but late responder, only directly at sites of necrosis caused by *Cercospora* infection [54].

Interestingly, only chitosan enhanced the expression of additional defense-related pathways by increasing levels of PAL (phytoalexin synthesis) and oxidative stress-related genes (GST and DMRL) as defense mechanisms. In this context, it is important to mention that chitin-specific receptors in plants recognize chitin and its derivatives when they are released by interaction with fungal pathogens, which then elicit defense responses. Accordingly, a chitin-based treatment activates a defense mechanism in plants, mimicking the attack by a pathogen [55]. Chitin elicitor binding proteins (CEBiP) isolated from several crops affect the chitosan-responsive differential gene expression profile and, thus, defense responses directly [56]. These mainly include phytoalexin biosynthesis (hypersensitive response, HR), which involves the participation of enzymes of phenylpropanoid metabolism, i.e., phenylalanine ammonia lyase (PAL) and chalcone synthase (CHS) [9]. PAL is involved in the synthesis of aromatic secondary metabolites e.g., phytoalexins, and may thus directly contribute to disease resistance in hypersensitive response (HR) [9,13]. The induction of the PAL gene is an indicator of induced cell wall thickening and might also hint at the production of antifungal compounds, such as phytoalexins. In order to facilitate the infestation, *C. beticola* directly suppresses the PAL expression in the first step of the infection [57]. Rivera-Méndez et al. [12] recently found evidence of the suppression of PAL expression by a fungal pathogen in bulbous plants (*Allium cepa* L.) on the first day of infection with *Sclerotium cepivorum*. In addition, pretreatment or simultaneous treatment with a *T. asperellum* isolate (BCC1) induced PAL expression, even if the induction was not stable over a long period of time. Our experiments provide evidence that the systemic resistance induced by chitosan treatment leads to increased PAL expression, which could significantly reduce the incidence of CLS, by cell wall thickening and/or phytoalexin production. Notably, overexpression of PAL in tobacco resulted in increased production of the phenylpropanoid compound chlorogenic acid and reduced susceptibility to infection with *Cercospora nicotianae* [58,59].

Cercosporin, produced by *C. beticola*, when exposed to light leads to the production of reactive oxygen species (ROS). ROS, in turn, are used by *C. beticola* to break down plant cell membranes [60]. The increase in antioxidant gene expression (increased GST and DMRL transcription) observed in our experiments is a strong indication for detoxification of the generated ROS in *B. vulgaris* treated with HMW chitosan. In addition, peroxidases can use ROS to increase lignin and, thus, cell wall thickness by oxidizing phenolic compounds [61].

Seed coatings provide protection during the first stages of germination and seedling development, where the seeds and seedlings are most vulnerable and may not be fully capable of protecting themselves from pathogens. Above all, chitosan, with its film-forming, viscous properties [21], is also ideally suited for coating. Using radio-labelled chitosan, it has further been shown that chitosan can be translocated to the emerging seedling [62].

Positive effects of chitosan on germination, growth, and plant health have been described previously for several plants; however, to our knowledge, *B. vulgaris* seed coating with chitosan has not been investigated so far [4]. In a filter paper assay with coated *B. vulgaris* seeds, we observed a positive impact of HMW chitosan on germination onset and germination efficiency compared to the control condition. After 5 days in the dark, an increase in total germination of 15% compared to the control group was observed. We speculate that a long-lasting effect, resulting in increased beet development and weight, might positively affect sugar yield. Despite the strong growth promotion, we could not find any differences in the expression profile of defense genes of the seedlings. Expression analysis of a different set of genes, such as those related to nitrogen metabolism, or to regulation of developmental hormones, including abscisic acid and auxin, could help to identify the underlying mechanism of growth promotion in seedling development. Interestingly, different developmental stages and tissues might benefit differently from the seed coating with chitosan, but resulted in increased over all germination efficiency and perhaps general development.

Interestingly, when the growth medium was supplemented with 1 mg mL^−1^ of chitosan, instead of coating with HMW chitosan, all screened defense-related genes were highly upregulated in the roots, but only PR-3 expression was induced in the shoots of the seedlings, suggesting induced systemic resistance in the above-ground parts. A stronger root hair development and a shorter stunted growth of the radicle were observed in these seedlings (data not shown). The shorter roots might be a tradeoff for the stronger root hair formation, which might in turn increase nutrient mobilization and growth promotion of the seedling in later growth periods. *Arabidopsis* root colonization by *P. indica* resulted in a stunted but highly branched root system, which was probably mediated by low amounts of auxins produced by *P. indica* [6].

*Trichoderma* spp. are mainly used as pre-planting applications for seed coatings and, similar to our data with chitosan coating, have been shown to improve germination of a variety of plants, their vitality index, and defense mechanism [5]. The seed coating of *B. vulgaris* with *T. atroviride* spores in our experiments had a positive influence on the defense-related gene expression. Some *Trichoderma* species are able to penetrate the epidermis and establish long-lasting colonization of the root surfaces, which, in turn, leads to changes in the plants metabolism and expression profile [5]. We noticed that *T. atroviride* had established colonization on the roots and lower parts of the hypocotyl of the seedlings, while no growth was observed on the shoots surface (data not shown). Notably, we detected an increased expression of *epl1*, an indicator for communication with the plant [19], in *T. atroviride* isolated from these roots. EPL1 is highly similar to cerato-platanins, which have been implicated in plant pathogenesis and in the elicitation of plant defense responses. In addition, the whole set of investigated defense-related genes, except for PAL, was highly upregulated in roots and shoots of *B. vulgaris*, by the presence of the BCA, suggesting activation of the defense-related machinery against pathogens, but reduced lignification to facilitate entry of the BCA. This might be due to the stress generated by the strong colonization of *T. atroviride* of the roots in the sterile culture.

Our data suggest that CHSN is an excellent candidate to replace or assist common fungicides by promoting seedling germination, growth, and plant health, as well as inducing multiple systemic resistance. We were able to show how chitosan, in particular, has a biocidal, immune-stimulating effect, while *T. atroviride* spores mainly induced expression of a stress-related defense gene in *B. vulgaris.* The incidence of CLS disease was effectively reduced by priming with chitosan and, to a lesser extent, by *T. atroviride* spores in foliar application, but both active substances revealed a pronounced insect-repellent effect. We assume a synergistic effect of *T. atroviride* and, in particular, of HMW chitosan on the inhibition of relevant *B. vulgaris* pathogens due to an enhanced expression of mycoparasitism-associated enzymes. Both agents could also act synergistically across different signaling pathways, which could be of great advantage for their combinatorial (and, thus, complementary) application. The mechanisms of the perception of chitosan by plants and their induced resistance have not yet been elucidated, but it is known that the expression of RLK genes is induced in various plant species and the mitogen-activated protein kinase (MAPK or MPK) is activated. In addition, the chitin receptor AtCERK1 binds weakly to chitosan (reviewed in [63]). In the signal transduction mediated by the plant defense system, MAPK cascade signal networks also play an essential role in establishing resistance to pathogen defense, in particular the MPK3, MPK4, and MPK6 cascades [64]. It would, therefore, make sense to identify the differences with greater precision.

Notably, a combined use of CHSN and *Trichoderma* spores also demands caution, as also only a small mutual inhibitory effect of both agents could affect disease efficacy. To this end, data on experiments in dual applications need to be considered. Nevertheless, an alternating application appears to be more successful towards a possible complementary function in signaling pathways. Moreover, concentration and strong-sensitivity fluctuations of the beneficial soil microbiome should be considered and empirically determined in further *in vitro* assays as major decisive factors for plant treatments.

In summary, it is thus conceivable that spraying *Trichoderma* spp., in combination with exogenous chitosan, may be used as a powerful tool to enable crop control. Taking into account the fact that a frequent change in fungicides can bypass pathogen resistance but trigger plant defense reactions, the combined or alternating applications of these BCAs would be a fascinating new tool, which we will investigate in detail in further studies.

## 5. Conclusions

Our results showed that the coating of *B. vulgaris* seeds and preventive foliar application with chitosan promotes growth, resistance, and defense gene expression and reduces CLS incidence. The first results for a combination of HMW chitosan and *T. atroviride* in an *in vitro* confrontation assay highlight their great potential, putatively setting the stage for a future trend towards advances for a combined use in biocontrol applications.

## Figures and Tables

**Figure 1 jof-08-00137-f001:**
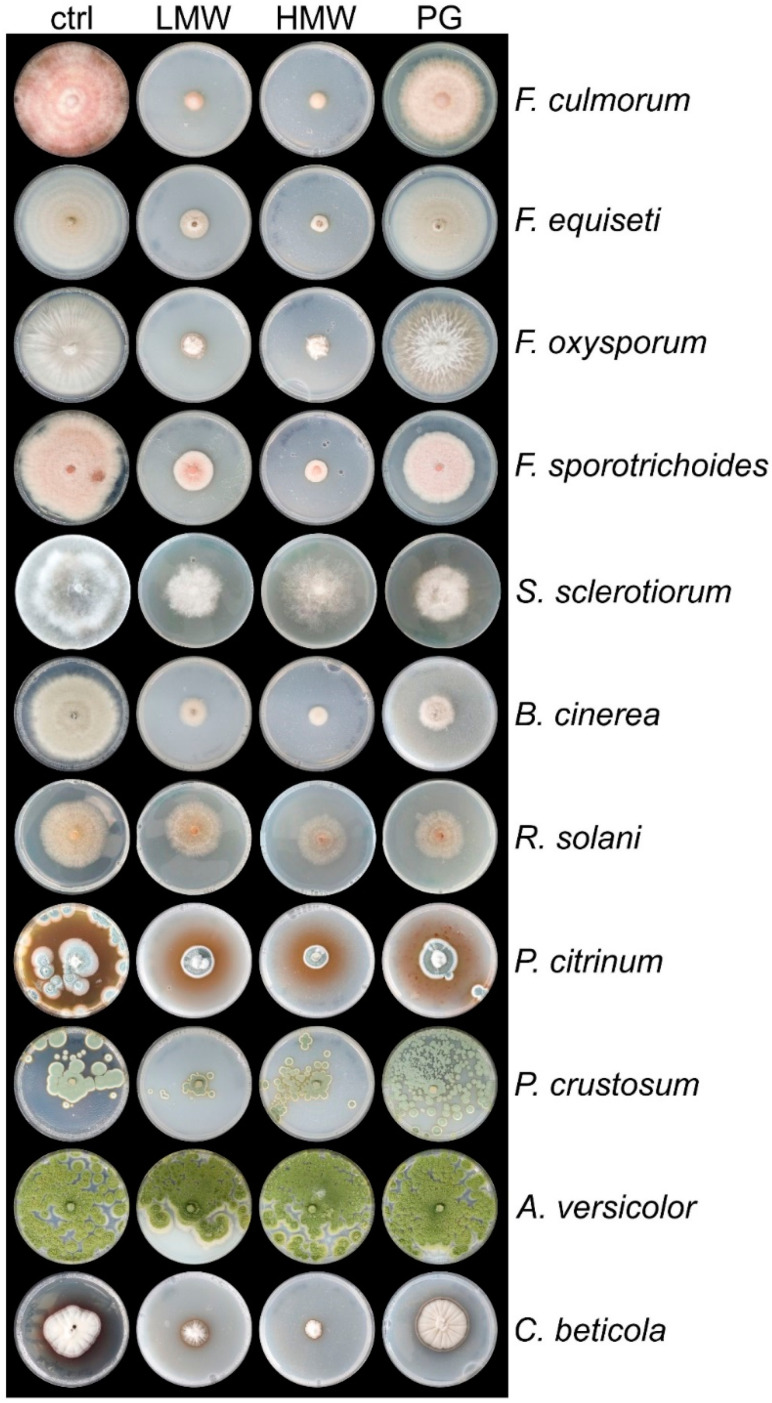
Growth of plant pathogens is strongly impaired by chitosan. Representative pictures of the indicated pathogens and *T. atroviride* were cultured on potato dextrose agar (PDA) and supplemented with 2 mg mL^−1^ of chitosan with different grades: LMW, low molecular weight; HMW, high molecular weight; PG, practical grade, and compared to PDA (control). Pictures were taken when the cultures on PDA reached the plate rim or started to sporulate.

**Figure 2 jof-08-00137-f002:**
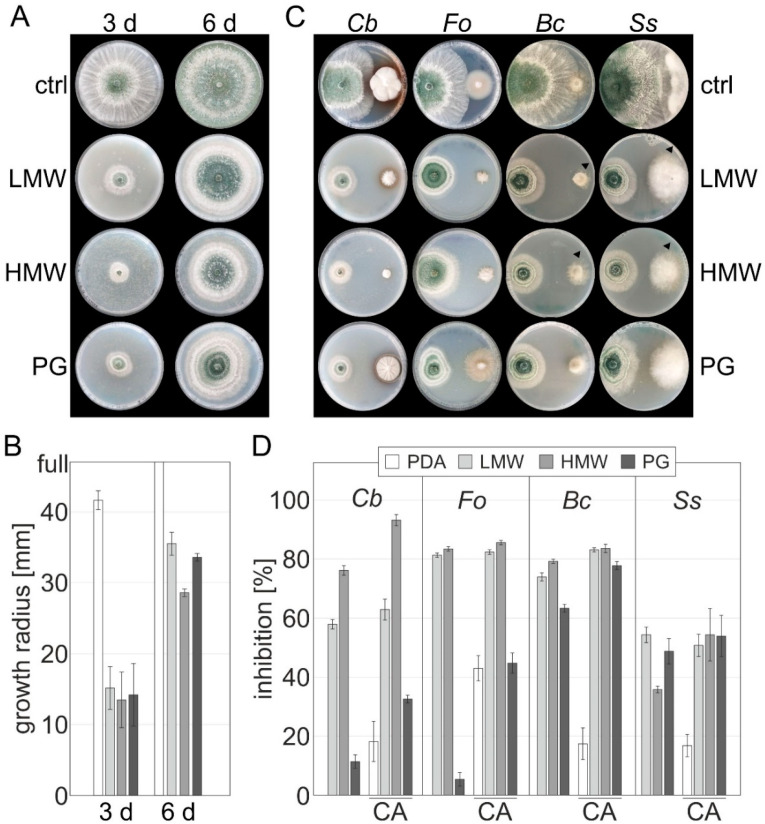
Synergistic effect of *T. atroviride* and chitosans on the growth of selected pathogens. Strains were cultured on (PDA, control) or supplemented with 2 mg mL^−1^ of chitosan: LMW, low molecular weight; HMW, high molecular weight; PG, practical grade. (**A**) Growth of *T. atroviride* after 3 (3 d) and 6 days (6 d). (**B**) Colonial extension of *T. atroviride* measured in mm after 3 (3 d) and 6 days (6 d). (**C**) Agar plugs from the indicated pathogens were placed approximately 50 mm opposite to *T. atroviride* in a dual confrontation assay on plates. (**D**) The growth radius of all pathogens and *T. atroviride* was recorded and the inhibition in percent was calculated by comparing growth on PDA without chitosan and without a challenging opponent (w/o *T. atroviride*, Appendix A) with the respective conditions. CA, confrontation assay; standard deviations are indicated for (**B**,**D**); *Cb*, *C. beticola*; *Fo, F. oxysporum*; *Bc, B. cinerea; Ss*, *S. sclerotiorum*.

**Figure 3 jof-08-00137-f003:**
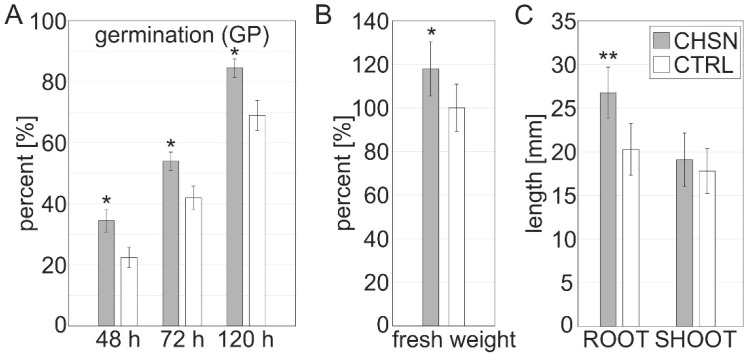
Germination and seedling growth is promoted by HMW chitosan. Germination percentage (GP) from Table 1 (**A**), fresh weight (**B**), and root and shoot length (**C**) of seedlings germinated for 10 days in the dark from seeds coated with 10 mg mL^−1^ of HMW chitosan or acidified water as control condition in a filter paper assay. * indicates significance at *p* < 0.05 and ** indicates significance at *p* < 0.01. *n* = 100 seeds per condition in two independent experiments.

**Figure 4 jof-08-00137-f004:**
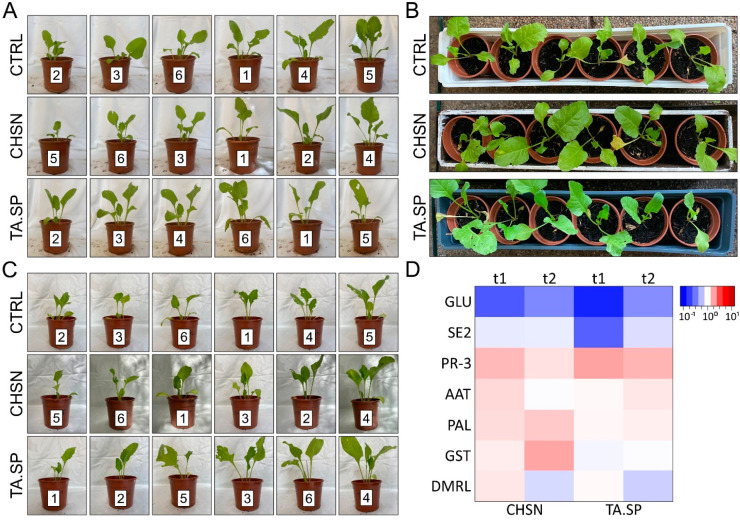
Leaf application of both HMW chitosan and *T. atroviride* spores are beneficial for growth promotion of young *B. vulgaris* plants. (**A**–**C**) *B. vulgaris* was grown in a greenhouse from May to August 2021 without additional heating. At 37 and 57 days, the leaves of the plants were either sprayed with deionized water pH 4.5 (control), 1 mg mL^−1^ of high-molecular-weight chitosan solution, pH 4.5 (HMW), or a freshly prepared *T. atroviride* 10^4^ spores mL^−1^ solution in deionized water (TA.SP). Twenty days after the first treatment (**A**,**B**) and thirteen days after the second treatment (**C**), the plants were photographed and arranged according to size for better comparison. (**D**) At the same time as shown in (**A**,**C**), a leaf from each plant in comparable size was harvested for RNA extraction and expression analysis. The data from all six plants per condition were combined and compared to the control condition for each treatment. The data are presented in a heatmap where red indicates upregulation and blue downregulation of expression. GLU, SE2, and PR-3 indicate expression of genes related to pathogenesis; AAT and PAL are related to plant growth and cell wall strengthening; and GST and DMRL are related to the oxidative burst. See Appendix A for statistical significance. *n* = 6 biological replicates and 3 technical replicates.

**Figure 5 jof-08-00137-f005:**
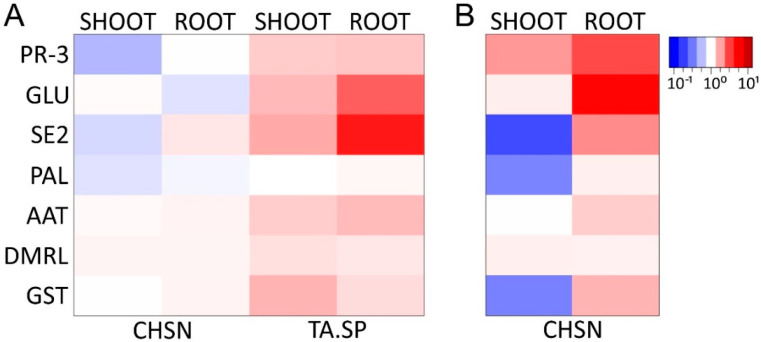
Strong upregulation of indicator genes in the roots and hypocotyl of seedlings grown on chitosan supplemented medium. *B. vulgaris* roots and shoots from 10-day-old seedlings from *in vitro* cultivation in a circadian rhythm were harvested for RNA extraction and expression analysis. (**A**) Seedlings were grown from seeds coated with 10 mg mL^−1^ of HMW chitosan or *T. atroviride* 10^4^ spores mL^−1^ or uncoated seeds. For *in vitro* cultivation, phytagel supplemented with sucrose was used. (**B**) Uncoated seeds were grown in medium supplemented with sucrose and 1 mg mL^−1^ of HMW chitosan. The expression data are presented in a heatmap where red indicates an upregulation and blue a downregulation of expression. See Appendix A for statistical significance. *n* = 5–6 biological replicates and 3 technical replicates.

**Figure 6 jof-08-00137-f006:**
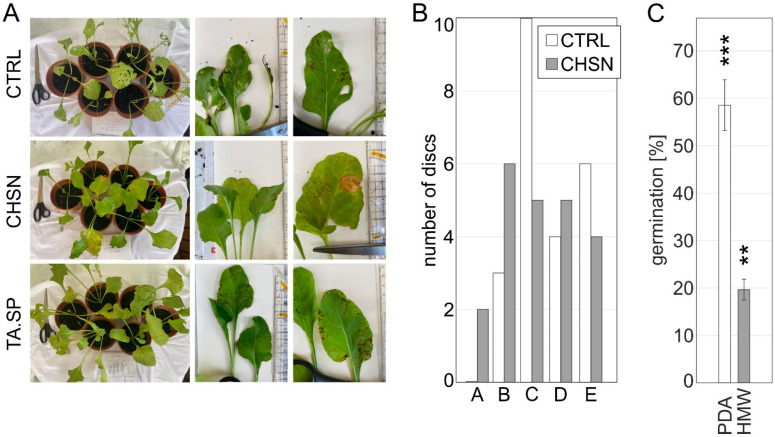
*B. vulgaris* young plants primed with HMW chitosan or *T. atroviride* spores show systemic resistance toward CLS from *C. beticola*. (**A**) *B. vulgaris* in a greenhouse experiment was infected with 10^4^ spores mL^−1^ from *C. beticola* seven days after the third foliar treatment with either 1 mg mL^−1^ of high-molecular-weight chitosan solution pH 4.5 (HMW), *T. atroviride* 10^4^ spores (TA.SP) or deionized water pH 4.5 (control). Photographs of the plants and leaves were taken 14 days post-infection to estimate CLS severity and frass (see Table 2). (**B**) Leaf disc assay to estimate disease severity of CLS disease. Leaves from *B. vulgaris* plants were sprayed 24 h before assay start with 1 mg mL^−1^ of HMW chitosan (CHSN) or deionized, acidified water pH 4.5 (control). 10 mm leaf discs were cut from these leaves, placed on water agar, and sprayed with 10^4^ spores mL^−1^ *C. beticola*. After 14 days, the development of CLS was recorded and categorized per leaf disc (*n* = 22 discs for CHSN and 23 discs for the control). The number of discs falling into each category is given. Category A = 0 CLS, B ≤ 5 CLS, C = 6–10 CLS, D = 11–20 CLS, and E > 20 CLS. (**C**) Germination efficiency of *C. beticola* compared to total amount of spores spread on PDA or PDA supplemented with 1 mg ml^−1^ of HMW chitosan (HMW). ** and *** indicate *p* < 0.01 and *p* < 0.001 (Poisson), respectively.

**Table 1 jof-08-00137-t001:** *B. vulgaris* germination.

Treatment	GP (t = 5 d)	GE (t = 48 h)	GE (t = 72 h)	GE (t = 120 h)	GI (t = 5 d)
Control	69.0 ± 4.9	11.25 ± 1.6	14 ± 1.3	13.8 ± 1.0	13.02 ± 1.29
CHSN	84.5 * ± 3.0	17.25 * ± 1.8	18 * ± 1.0	16.9 * ± 0.6	17.38 * ± 1.14

Values expressed as mean ± standard error. * Data are significantly different at *p* < 0.05. *n* = 100 in 2 technical replicates. GP, germination percentage (germination after 5 days (%)); GE, germination capacity (germination efficiency (%) per day after t = hours); GI, germination index (germination mean (%) per day over 5 days).

**Table 2 jof-08-00137-t002:** Disease severity of *B. vulgaris* after 2 weeks of infestation with *C. beticola*.

Treatment	Disease Incidence (%)	Visual Assessment/ Dark Leave Spots	Visual Assessment/ Frass Incidence (%)
Control	78.79	severe/skeletal lesions	33.33
0.1% HMW	31.43	moderate/light lesions	14.29
*T. atroviride* spores	51.72	moderate/dark lesions	20.00

## Data Availability

Not applicable.

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
