# Peer review of "The Multilateral Efficacy of Chitosan and Trichoderma on Sugar Beet"

_jof, 2022, doi:10.3390/jof8020137_

Round 1

Reviewer 1 Report

The multilateral efficacy of chitosan and Trichoderma on sugar beet. In this manuscript, the research method is too simple and the research data are simple statistical analysis. Also, the research content is not innovative enough and the research is not explore internal mechanism of chitosan and Trichoderma on sugar beet. Therefore, it is suggested to reject the manuscript and encourage resubmission after modify text and supplement data.

Detailed Comments:

  1. In this manuscript, the 1456 lines and 133 references were provided by the author. Obviously, the manuscript does not conform to the specifications of scientific paper writing and not concise and comprehensive.
  2. Trichoderma atroviride or Chitosan was single used to reflect positively growth-promotion of Beta vulgaris (Fig.4 and Fig.5). There is no data about combined treatments of atroviride and chitosans to Beta vulgaris.
  3. The article theme is not clear. atroviride and chitosans were recommended used in combination or alone? If alone usage was recommened, compared with T. atroviride and chitosans fungicidal effects is not significant. If combination was recommened, it is no adequately experiment data.
  4. Supplement legend for the Fig.2B and Fig.6C.
  5. Line 363-364: spp
  6. Line 390: Replaced the “.” after “12 days” with “,”.

Author Response

Response to Reviewer 1 Comments

Point 1: Detailed Comments:

Comments and Suggestions for Authors

The multilateral efficacy of chitosan and Trichoderma on sugar beet. In this manuscript, the research method is too simple and the research data are simple statistical analysis. Also, the research content is not innovative enough and the research is not explore internal mechanism of chitosan and Trichoderma on sugar beet. Therefore, it is suggested to reject the manuscript and encourage resubmission after modify text and supplement data.

We are pleased to present the revised manuscript and our point-to-point answers to the reviewers’ suggestions. The revised manuscript is now shorter, more concise and the excess of references was reduced considerably and complemented, as suggested by the reviewers. We also added supplemental data as requested by the reviewers.

General Response 1: Concerning the length of the manuscript, the revised version of the manuscript is considerably shorter and more concise now, and should thus fit the JoF guidelines.

Concerning the innovative character: We show clearly and comprehensibly in the manuscript, how chitosan and T. atroviride spores positively affect plant growth, health and disease resistance of B. vulgaris, which has not been shown so far. Our results are based on complete datasets with significant data (number of experiments, number of biological replicates and statistical analysis conform to scientific standards as can be seen in methods and results section). We also present a first in vitro assay that combines T. atroviride and chitosan, which leads to a synergistic growth reduction of important sugar beet pathogens. This experiment clearly shows the high potential for a combined use of the two agents and is clearly an innovative approach. In the discussion, we reflect on the difficulties and possible drawbacks, that could arise by a combined treatment on the field and therefore it should be clear that this approach needs to be followed with caution and reconsideration of more experiments, and could thus not be included already in the current manuscript. Nevertheless, it is important to present our findings to the broad scientific community and think that JoF is the right journal for this.

Response 1:

With our work, we have made the first attempt to enter a hitherto insufficiently explored field of research. Hence, it is important to start with experimental approaches, which demonstrate the efficacy of both agents on plant growth promotion and disease resistance before investigation and demonstrating underlying difficult intramolecular mechanisms. Nevertheless, we provide first insights in the intracellular mechanisms of disease resistance in sugar beet, by analyzing the expression profile of marker genes for induced systemic resistance, pathogenesis related factors, reactive oxygen detoxification and defense related genes in young plants and seedlings. Thus, we already provided first data on internal mechanisms of chitosan and Trichoderma on sugar beet.

The innovative content relates to the enormous effects that could be achieved phenotypically by both agents. Further investigation of chitosan and Trichoderma treatments on intracellular mechanisms, while important and interesting, goes beyond these initial study goals and will be carried out in the near future.

Our work is indeed very important as it demonstrates potent biostimulating and resistance-enhancing effects on an important crop. This has implications for basic research and general applications. Here, we would like to point out that preliminary or small data sets should not be confused with less innovation.

Furthermore, our findings are very important and independent of combined or individual use for future applications, with the presented initial data favoring a reason for a trend towards alternate use, as discussed in the manuscript.

We agree with the reviewer: The number of citations as well as the length have been reduced and the structure of the manuscript has been improved and the "review character" has been adapted to the requirements of a scientific article and to JOF standards.

Detailed Comments:

Point 2: In this manuscript, the 1456 lines and 133 references were provided by the author. Obviously, the manuscript does not conform to the specifications of scientific paper writing and not concise and comprehensive.

Response 2:

The manuscript was considerably shortened and the number of references reduced to 105. We worked on comprehensibility, following also suggestions by reviewers 2 and 3. The introduction is now more concise. The results present the data without interpretation and hypotheses and conclusions are presented in a clear way in the discussion and include now also references, that just were released on this topic, as suggested by reviewer 2. We are confident, that the manuscript in the current form will now conform to the specifications of JoF and will be interesting to a broad scientific community.

Point 3: Trichoderma atroviride or Chitosan was single used to reflect positively growth-promotion of Beta vulgaris (Fig.4 and Fig.5). There is no data about combined treatments of atroviride and chitosans to Beta vulgaris.

Response 3:

Data on combination are demonstrated as first in vitro assays to estimate the antagonism of the Trichoderma and Chitosan (Figure 1). In planta, experiments will be carried out next season. Since, to date, no data is available on the use of neither chitosan nor T. atroviride for growth promotion or preventive disease treatment in sugar beet, the data on the single use of both agents, which we provide in this manuscript, was a prerequisite for further experiments. With our set of experiments, we could show for the first time, that T. atroviride and chitosan are potent agents for use in B. vulgaris cultivation. We also provide convincing data on a combined use of T. atroviride and chitosan in a first in vitro confrontation assay involving the most relevant plant pathogens for sugar beet cultivation. In the discussion, we outline very clearly, which steps are planned for a combined use and what difficulties and drawbacks could be faced on field.  

Point 4: The article theme is not clear. atroviride and chitosans were recommended used in combination or alone? If alone usage was recommened, compared with T. atroviride and chitosans fungicidal effects is not significant. If combination was recommened, it is no adequately experiment data.

Response 4:

The title was changed, due also to suggestions of reviewer 3. The title should now more clearly describe our findings and fit to the data presented in the manuscript. We apologize for any confusion concerning the single or combined use of chitosan and T. atroviride and also introduced minor changes in the abstract to clarify our goals and presented results.  Regarding the significance of our findings, we oppose to the view of reviewer 1, and are confident, that our data are indeed significant: The fungicidal effects of chitosan are obvious in the plate assays (Fig1); and the priming of the plants, with either chitosan or T. atroviride, resulted in both, significant growth promotion and resistance to CLS disease in the young B. vulgaris plants (Fig 4, 6 and Table 2). Moreover, the seedlings germination efficiency, wet weight and root length were significantly increased by seed treatment with chitosan (Fig 3 and Table 1).

Point 5: Supplement legend for the Fig.2B and Fig.6C.

Response 5:

We provided supplemental data for Figure 2C and D, i.e. growth of the pathogenic fungi on PDA, and on PDA supplemented with chitosans, but without confrontation with T atroviride, corresponding control condition to the confrontation assays shown in Figure 2C. We hope this is consistent with what Reviewer 1 suggested as Figures 2A and 2B, providing information on T. atroviride growth on chitosan, are complete and supplementary data are now available here. We also provide supplemental information for Figure 6C now, with pictures of the colonies formed from single spores of C. beticola.

Point 6: Line 363-364: spp

Response 6: corrected

Point 7: Line 390: Replaced the “.” after “12 days” with “,”.

Response 7: We did not replace “.” with “,” since the sentences cannot be connected.

Reviewer 2 Report

The present manuscript submitted for review mainly includes information about the ability of a strain of Trichoderma atroviride and chitosan to promote growth and induce defense mechanisms in Beta vulgaris against the pathogen Cercospora beticola.

In general terms the topic of this article is in line with this special issue. The work does not raise any scientific reservation about methodology and results reported. However, I have major concerns in the current state of this manuscript.

Main concerns:

1- There are conceptual errors. In this sense, terms like “pests” and “fungal diseases” or “pesticides” and “fungicides” should not be exchanged. Several sentences which are listed below would be rewritten in the sections “Introduction“ and “Discussion”.

2- Microorganism names are confusingly used throughout the manuscript (the complete name should be used only the first time which is quoted).

3- Acronyms are confusingly used throughout the manuscript. i.e. Cercospora leaf spot disease (CLS) is used in the Abstract but Cercospora leaf spot (CLS) is used in the Introduction. Please, check this and the others.

4- Results should be concisely summarized. Many paragraphs in this section should be considered as part of Introduction, M&M or Discussion.

5- The section “References” must be carefully checked since different formats were used. When authors refer the journal names (abbreviated or not, …), the title of articles (capital letters or not), and the italics for microorganism names a little coherence is needed.

6- More adequate references are needed in some sentences to illustrate the work. It is especially important when review articles are cited. i.e. Poveda et al., 2021, this author has published a review about Trichoderma against pests, in my opinion it is not a fine revision, and it must not be used in the context of fungal disease biocontrol. Moreover, I consider that other references should be used here (i.e. Mukherjee et al (2022). “Mycoparasitism as a mechanism of Trichoderma-mediated suppression of plant diseases” seems adequate, although this one was probably not available for the authors a few weeks ago.  

7- Many references should be removed to avoid confusion (i.e. Nicolás et al., 2014; Sánchez-Montesinos et al. 2019; Stoppacher et al., 2010; Daniel and Guest, 2005; Valenzeno and Pooler, 1987; Coppola et al. 2019). Livak and Schmittgen, 2001 should be removed when Pfaffl et al has been also used.

Thus, the following points should be considered or changed:

Abstract:

- Line 20: Cercospora leaf spot disease (CLS) (Cercospora without italics, it is a disease name) and this should be also changed in keywords; Beta vulgaris (B. vulgaris should be used)

- Line 21, “pesticides” should be changed by “fungicides”.

Introduction:

- Line 33: Better use “microorganism”

- Line 49: This sentence should be rewritten as above indicated in point 1.

- Lines 60-61: Actinomycetes, Mycorrhiza (better mycorrhizal fungi) and Rhizobacteria were used as generically names and they should be indicated without italics and without capital letters.

- Line 64. Here, a reference is needed. Illescas et al (Front. Plant Sci. 2020) could be adequate in this context.

- Lines 65, 84, 85, and … throughout the manuscript (spp. without italics)

- Lines 71 and 75, reference Poveda (2021) must be removed here and other parts of manuscript, as above indicated in point 6.

- Line 81: Rivera-Méndez et al (2020), working in greenhouse/field experiments, have related plant responses triggered by Trichoderma with PAL. This report is suggested in this paragraph since some results about the expression levels of PAL genes and PAL activity in Trichoderma-treated plants, with and without the pathogen, should be considered in different paragraphs of the “Discussion” section.

- Lines 81, and 149-158: More adequate references can be used here. (i.e. Pedrero-Méndez et al (Journal of Fungi, 2021) has reported in this Special Issue that a mixture of four Trichoderma strains by triggering antioxidant enzymatic activities in the plant increases wheat defenses against abiotic stress and yield grain. This article is now available for the authors.

- Lines 104, 119, 150, 159 (Use T. atroviride or T. harzianum as appropiate).

- Line 122: Cercospora leaf spot (CLS), check considering my above comment about the abstract.

- Line 135: Better remove “due to a CLS infection”.

- Line 136: CLS should be used here.

- Line 138: “pesticides” is not adequate. Maybe “these products” could be used.

- Line 155: Here, “pests” is not an adequate term.

- Line 160: B. vulgaris should be used.

- Line 170: Here, CLS should be used, as above indicated.

- Line 173: B. vulgaris should be used.

M&M:

- Lines 181-182: “(teleomorph: Hypocrea atroviridis, (https://mycocosm.jgi.doe.gov/Triat1/Triat1.home.html)” should be removed. Strain used in this work is very known. Nowdays it is not recommended to use the teleomorh term in BCA works.

- Lines 181-193: Use the name of microorganisms and plant as above pointed.

- Lines 211-212: This sentence should be rewritten and shortened. “… PDA petri plate with 2 mg ml-1 chitosan” is enough.

- Lines 223-224: This sentence seems incomplete and it should be improved.

- Lines 230, 231, 240, 284, 338, 364, 365, 368, 388, 397, … etc. Please, check name of microorganisms and plant, this is not a task for a reviewer.

Results:

Please, only results obtained in the present work should be included in this section. Many sentences and paragraphs must be removed or moved to other sections.

- Lines 362-365: Names of plants and fungi should be corrected. Moreover, “spp.” must be indicated without italics.

- Lines 356-360: Remove.

- Lines 360-360: This paragraph should be summarized, and only results should be included. Moreover, in my opinion, many sentences can be removed. (Lines 360-366, M&M; Lines 366-380 are not results).

- Lines 394, and 398-399: Remove here, they should be moved to the Discussion.

- Line 401: “agrobiologically important”, remove it. It was not demonstrated in this work.

- Line 414: “severe”, remove.

- Lines 418-422: remove.

- Lines 434-440: remove (M&M).

- Lines 446-449: remove (Discussion)

- Lines 452-454: remove (Discussion)

- Lines 467-470: This sentence should be rewritten or removed.

- Lines 471-477: remove (Discussion)

- Lines 489-495: remove (hypothesis/Discussion)

- Lines 512-519: remove (Introduction)

- Line 540: “ability” should be removed.

- Table 1: All data are with * in this table. Is it correct?

- Lines 556-571: remove (Introduction)

- Line 583: “regulation”

- Lines 583-584: Italics should be used for genes, or this sentence should be rewritten.

- Lines 599-602: remove (Discussion).

- Lines 606-610: remove (Introduction).

- Lines 649-660: remove (Introduction).

- Lines 662-671: remove (Discussion).

- Lines 682: “To our surprise”, remove.

- Lines 687-691: remove (Discussion).

- Lines 718-728: remove (Discussion).

- Lines 731-742: remove (Introduction/Discussion).

- Lines 757-759: remove (Discussion).

- Lines 761: Authors use Trichoderma atroviride and Beta vulgaris on page 18.

- Lines 766-768: remove (Discussion).

- Lines 769-770: remove (Conclusions).

- Lines 800-808: remove (Discussion).

Discussion:

- Lines 823-828: remove. In the present work, authors have studied Trichoderma and chitosan as biofungicide. Thus, pests should be left out.

- Line 830: “pesticides” should be changed by “fungicides”.

- Microorganisms and plants must be appropriately named throughout this section (i.e. lines 831, 832, 836, 837, …) and "spp." without italics.

- Abbreviation must be adequately used (i.e. lines 840, 842, 933, … COS, PG, CLS, … They were previously indicated).

- Lines 918-919: As above indicated. Several plant mechanisms are triggered by the application of Trichoderma, particularly against abiotic stresses, Pedrero-Méndez et al (Journal of Fungi, 2021) has reported in this Special Issue that a mixture of four Trichoderma strains increases wheat defenses against abiotic stress and yield grain by triggering antioxidant enzymatic activities in the plant. Thus, most adequate articles that these cited by authors are available.

-Lines 972-974: These results should be discussed considering those previously reported by Rivera-Mendez et al (Biological Control, 2020), particularly those about enhanced expression of levels of PAL in addition to increasing PAL activity as consequence of Trichoderma treatment.

- Lines 987-989: Changes caused by the pathogen Sclerotinia in PAL expression levels in onion plants, described in Rivera-Méndez et al (2020), should be discussed here.

- Line 1064: “pesticides” should be changed by “fungicides”.

- Line 1091: “pesticides” should be changed by “fungicides”.

References:

-As above indicated (points 5, 6 and 7).

Author Response

Response to Reviewer 2 Comments

Point 0: The present manuscript submitted for review mainly includes information about the ability of a strain of Trichoderma atroviride and chitosan to promote growth and induce defense mechanisms in Beta vulgaris against the pathogen Cercospora beticola.

In general terms the topic of this article is in line with this special issue. The work does not raise any scientific reservation about methodology and results reported. However, I have major concerns in the current state of this manuscript.

Response 0: We would like to thank the reviewer for the overall positive evaluation, helpful comments and expert opinion on our work as well as for the detailed feedback and thus efforts he/she made to improve the strucutre and terminology on the current state of the manuscript. The suggestions were evaluated enclosed to the manucript, and amended in the the text. We believe that the comments have been properly addressed, and that the changes made in the manuscript helped to significantly improve its overall quality and readabilty. All changes are comprehensable due to track change mode – references changed are higlighted. We are convinced that the sometimes massive loss of context will not degrade the scientific impact, but that the shortening of some very detailed passages will enable a more concise presentation of the results

Our point-to-point responses are listed below.

Main concerns:

Point 1: 1- There are conceptual errors. In this sense, terms like “pests” and “fungal diseases” or “pesticides” and “fungicides” should not be exchanged. Several sentences which are listed below would be rewritten in the sections “Introduction“ and “Discussion”.

Response 1:  We agree with the comments and apologize for any  conceptual errors in the text. We also apologize for any terminology errors – and for use of wrong terms in any cases.  We have assessed the manuscript and corrected errors in wrong use of terms.

Point 2: 2- Microorganism names are confusingly used throughout the manuscript (the complete name should be used only the first time which is quoted).

Response 2: The entire manuscript has been reviewed and corrected for this

Point 3: 3- Acronyms are confusingly used throughout the manuscript. i.e. Cercospora leaf spot disease (CLS) is used in the Abstract but Cercospora leaf spot (CLS) is used in the Introduction. Please, check this and the others.

Response 3: The entire manuscript was checked and changed either into Cercospora leaf spot (CLS) disease  or CLS where appropriate.

Point 4: 4- Results should be concisely summarized. Many paragraphs in this section should be considered as part of Introduction, M&M or Discussion.

Response 4: We agree with the comment and have checked the correct embedding of the text as this was also an important concern of reviewer 3 and have improved the manuscript in this regard.

Point 5: 5- The section “References” must be carefully checked since different formats were used. When authors refer the journal names (abbreviated or not, …), the title of articles (capital letters or not), and the italics for microorganism names a little coherence is needed.

Response 5: We adjusted the output style to JoF standards - and corrected the capital letters manually or not, as this is always a difficult story with any citation management software.

Point 6: 6- More adequate references are needed in some sentences to illustrate the work. It is especially important when review articles are cited. i.e. Poveda et al., 2021, this author has published a review about Trichoderma against pests, in my opinion it is not a fine revision, and it must not be used in the context of fungal disease biocontrol. Moreover, I consider that other references should be used here (i.e. Mukherjee et al (2022). “Mycoparasitism as a mechanism of Trichoderma-mediated suppression of plant diseases” seems adequate, although this one was probably not available for the authors a few weeks ago.  

Response 6: Of course, we only found out about this excellent review article just prior to our submission and were unsure as it was just when we already finshed it. We changed the reference. Mukherjee et al., 2022 is indeed a very good review and we are happy to include it in the manuscript.

Point 7: 7- Many references should be removed to avoid confusion (i.e. Nicolás et al., 2014; Sánchez-Montesinos et al. 2019; Stoppacher et al., 2010; Daniel and Guest, 2005; Valenzeno and Pooler, 1987; Coppola et al. 2019). Livak and Schmittgen, 2001 should be removed when Pfaffl et al has been also used.

Response 7: We have removed or swapped out excess citations to avoid confusion - and changes have been highlighted in the manuscript text.

Thus, the following points should be considered or changed:

Abstract:

Point 8- Line 20: Cercospora leaf spot disease (CLS) (Cercospora without italics, it is a disease name) and this should be also changed in keywords; Beta vulgaris (B. vulgaris should be used)

Response 8: corrected

Point 9:- Line 21, “pesticides” should be changed by “fungicides”.

Response 9: corrected

Introduction:

Point 10: - Line 33: Better use “microorganism”

Response 10: corrected

Point 11:- Line 49: This sentence should be rewritten as above indicated in point 1.

Response 11: rephrased

Point 12:- Lines 60-61: Actinomycetes, Mycorrhiza (better mycorrhizal fungi) and Rhizobacteria were used as generically names and they should be indicated without italics and without capital letters.

Response 12: Due to recommendations by Reviewer 3, the sentence was deleted.

Point 13:- Line 64. Here, a reference is needed. Illescas et al (Front. Plant Sci. 2020) could be adequate in this context.

Response 13: Due to recommendations by Reviewer 3, the sentence was deleted.

Point 14:- Lines 65, 84, 85, and … throughout the manuscript (spp. without italics)

Response 14: corrected

Point 15:- Lines 71 and 75, reference Poveda (2021) must be removed here and other parts of manuscript, as above indicated in point 6.

Response 15: corrected

Point 16:- Line 81: Rivera-Méndez et al (2020), working in greenhouse/field experiments, have related plant responses triggered by Trichoderma with PAL. This report is suggested in this paragraph since some results about the expression levels of PAL genes and PAL activity in Trichoderma-treated plants, with and without the pathogen, should be considered in different paragraphs of the “Discussion” section.

Response 16: We included the citation in line 81 and added discussion concerning PAL expression and activity in lines: 1054-1059 in the version including the markups

Point 17:- - Lines 81, and 149-158: More adequate references can be used here. (i.e. Pedrero-Méndez et al (Journal of Fungi, 2021) has reported in this Special Issue that a mixture of four Trichoderma strains by triggering antioxidant enzymatic activities in the plant increases wheat defenses against abiotic stress and yield grain. This article is now available for the authors.

Response 17: The respective citation was included

Due to suggestions by reviewer 3 the section 149-158 was deleted, but the reference was used in line 81.

Point 18: - Lines 104, 119, 150, 159 (Use T. atroviride or T. harzianum as appropiate).

Response 18: corrected

Point 19: - Line 122: Cercospora leaf spot (CLS), check considering my above comment about the abstract.

Response 19: corrected

Point 20:- Line 135: Better remove “due to a CLS infection”.

Response 20: removed

Point 21:- Line 136: CLS should be used here.

Response 21: corrected

Point 22:- Line 138: “pesticides” is not adequate. Maybe “these products” could be used.

Response 22: corrected

Point 23:- Line 155: Here, “pests” is not an adequate term.

Response 23: changed to “pathogens”

Point 24:- Line 160: B. vulgaris should be used.

Response 24: corrected

Point 25:- Line 170: Here, CLS should be used, as above indicated.

Response 25: corrected

Point 26:- Line 173: B. vulgaris should be used.

Response 26: corrected

M&M:

Point 27:- Lines 181-182: “(teleomorph: Hypocrea atroviridis, (https://mycocosm.jgi.doe.gov/Triat1/Triat1.home.html)” should be removed. Strain used in this work is very known. Nowdays it is not recommended to use the teleomorh term in BCA works.

Response 27: We removed this part of the sentence and the teleomorph term in line 181 and in the introduction, line 83.

Point 28: - Lines 181-193: Use the name of microorganisms and plant as above pointed.

Response 28: corrected, the full name is now given only once and then abbreviated.

Point 29:- Lines 211-212: This sentence should be rewritten and shortened. “… PDA petri plate with 2 mg ml-1 chitosan” is enough.

Response 29: corrected

Point 30:- Lines 223-224: This sentence seems incomplete and it should be improved.

Response 30: rephrased

Point 31:- Lines 230, 231, 240, 284, 338, 364, 365, 368, 388, 397, … etc. Please, check name of microorganisms and plant, this is not a task for a reviewer.

Response 31: corrected.

Results:

Point 32: Please, only results obtained in the present work should be included in this section. Many sentences and paragraphs must be removed or moved to other sections.

Response 32: We have now removed the suggested sentences and paragraphs or transferred them to the discussion or methods, as suggested.

Point 33:- Lines 362-365: Names of plants and fungi should be corrected. Moreover, “spp.” must be indicated without italics.

Response 33: Names of fungi were abbreviated. “spp.” is now without italics

Point 34:- Lines 356-360: Remove.

Response 34: removed

Point 35:- Lines 360-360: This paragraph should be summarized, and only results should be included. Moreover, in my opinion, many sentences can be removed. (Lines 360-366, M&M; Lines 366-380 are not results).

Response 35: Paragraph was summarized and sentences transferred to other sections were we considered it important.

Point 36:- Lines 394, and 398-399: Remove here, they should be moved to the Discussion.

Response 36: removed and transferred.

Point 37:- Line 401: “agrobiologically important”, remove it. It was not demonstrated in this work.

Response 37: removed

Point 38:- Line 414: “severe”, remove.

Response 38: removed

Point 39:- Lines 418-422: remove.

Response 39: removed

Point 40- Lines 434-440: remove (M&M).

Response 40: removed

Point 41:- Lines 446-449: remove (Discussion)

Response 41: removed and inserted in discussion

Point 42:- Lines 452-454: remove (Discussion)

Response 42: removed and inserted in discussion

Point 43:- Lines 467-470: This sentence should be rewritten or removed.

Response 43: removed

Point 43- Lines 471-477: remove (Discussion)

Response 43: removed and partially moved to the discussion section

Point 44:- Lines 489-495: remove (hypothesis/Discussion)

Response 44: removed

Point 45:- Lines 512-519: remove (Introduction)

Response 45: removed

Point 46:- Line 540: “ability” should be removed.

Response 46: removed

Point 47:- Table 1: All data are with * in this table. Is it correct?

Response 47: We removed * from the ctrl condition, since this might be confusing for the reader, but all data from CHSN-condition are significantly different from control condition

Point 48:- Lines 556-571: remove (Introduction)

Response 48: removed

Point 49:- Line 583: “regulation”

Response 49: corrected

Point 50:- Lines 583-584: Italics should be used for genes, or this sentence should be rewritten.

Response 50: rewritten

Point 51:- Lines 599-602: remove (Discussion).

Response 51: removed and used in discussion

Point 52:- Lines 606-610: remove (Introduction).

Response 52: removed

Point 53:- Lines 649-660: remove (Introduction).

Response 53: removed

Point 54:- Lines 662-671: remove (Discussion).

Response 54: removed/ rephrased

Point 55- Lines 682: “To our surprise”, remove.

Response 55: Deleted

Point 56:- Lines 687-691: remove (Discussion).

Response 56: removed

Point 57:- Lines 718-728: remove (Discussion).

Response 57: Lines 718-722 were transferred to discussion and lines 723-728, which contain results, were rephrased.

Point 58:- Lines 731-742: remove (Introduction/Discussion).

Response 58: removed and partially rephrased

Point 59:- Lines 757-759: remove (Discussion).

Response 59: removed

Point 60:- Lines 761: Authors use Trichoderma atroviride and Beta vulgaris on page 18.

Response 60: corrected

Point 61:- Lines 766-768: remove (Discussion).

Response 61: removed and transferred to discussion

Point 62:- Lines 769-770: remove (Conclusions).

Response 62: removed

Point 63:- Lines 800-808: remove (Discussion).

Response 63: Lines 802-808 were removed and lines 808-809 rephrased accordingly.

Discussion:

Point 63:- Lines 823-828: remove. In the present work, authors have studied Trichoderma and chitosan as biofungicide. Thus, pests should be left out.

Response 63: removed

Point 64:- Line 830: “pesticides” should be changed by “fungicides”.

Response 64: changed

Point 65:- Microorganisms and plants must be appropriately named throughout this section (i.e. lines 831, 832, 836, 837, …) and "spp." without italics.

Response 65: corrected

Point 66:- Abbreviation must be adequately used (i.e. lines 840, 842, 933, … COS, PG, CLS, … They were previously indicated).

Response 66: corrected

Point 67:- Lines 918-919: As above indicated. Several plant mechanisms are triggered by the application of Trichoderma, particularly against abiotic stresses, Pedrero-Méndez et al (Journal of Fungi, 2021) has reported in this Special Issue that a mixture of four Trichoderma strains increases wheat defenses against abiotic stress and yield grain by triggering antioxidant enzymatic activities in the plant. Thus, most adequate articles that these cited by authors are available.

Response 67: We included the references.

Point 68:-Lines 972-974: These results should be discussed considering those previously reported by Rivera-Mendez et al (Biological Control, 2020), particularly those about enhanced expression of levels of PAL in addition to increasing PAL activity as consequence of Trichoderma treatment.

- Lines 987-989: Changes caused by the pathogen Sclerotinia in PAL expression levels in onion plants, described in Rivera-Méndez et al (2020), should be discussed here.

Response 68: We discussed the results concerning PAL expression and activity and the changes caused by Sclerotium in onion plants, and included the reference.

Point 69- Line 1064: “pesticides” should be changed by “fungicides”.

Response 69: corrected

Point 70:- Line 1091: “pesticides” should be changed by “fungicides”.

Response 70: corrected

References:

Point 71-As above indicated (points 5, 6 and 7).

Response 71: correted and highlighted in the text

Reviewer 3 Report

The manuscript titled “The multilateral efficacy of chitosan and Trichoderma  on sugar beet” evaluates the biocontrol efficacy of combining Trichoderma atroviride and chitosan against the Cercospora leaf spot disease, but also as a bioestimulant. Even though the results presented by Kappel et al. are interesting in terms of the few studies researching this topic and the potential of applicability of these formulations, and they are well disccused, how these results are presented makes the reading quite difficult and confusing. In this particular, comments listed below might be related to either the inadequate description of the experimental assay or the lack of description at all. Besides these aspects that need to be addressed for clarity and readability, there are many issues regarding editing references or the use of abbreviations for scientific names of microorganisms (when to use them and where).

Minor/major comments:

  1. References format, number or not accurately used.

Authors should carefully review references in order to match the requirements recommended by the journal. As an example, journal articles should follow the following criteria: 1. Author 1, A.B.; Author 2, C.D. Title of the article. Abbreviated Journal Name Year, Volume, page range.”  The first reference listed by authors has the following format:

Badawy MEI, Rabea EI: A Biopolymer Chitosan and Its Derivatives as Promising Antimicrobial Agents against Plant Pathogens and Their Applications in Crop Protection. International Journal of Carbohydrate Chemistry 2011, 2011. All references need to be reviewed and edited.

The number of references seems excessive, for example the 6 references used to indicate the benefits of Trichoderma secondary metabolites (Lines 104-107), or references that are not necessary like reference number 63.

The reference number 18 (Fiorentino et al., 2018) related to foliar applications, does not indicate any experimental assay to this regard. When referring to T. atroviride (lines 82-85), at least two of the references (number 20 and 21) do not evaluated the capacity of this specie against those pathogens or any pathogen at all.

  1. Introduction

Introduction is excessively extent and could be shorted, for example when referring to the aim of this research study (from line 149 to line 178).

In addition:

  • Line 94: “which”, should be corrected.
  • Lines 97-100: DAMPs can also trigger defense responses in the plant-Trichoderma
  • Lines 100-103: this sentence, referring to cerato-platanins of pathogens (references 29 and 30) is included in a paragraph referring to the properties of Trichoderma. There are many articles that describes these proteins like the well-known SM1 elicitor that might fit better that the current sentence. Also, lines 103-104 refer to EPL1 as a continuation of the description of the properties.
  • Lines 138-139: reference number 20 is not related to what the sentence is indicating (fungal resistances).
  • Line 155: “growing resistance of pests” or plant diseases?
  • Line 170-171: CLS abbreviation.

  1. Materials and Methods.

Titles of this section should be more descriptive. As an example, cultivation conditions section (line 205) includes preparation of PDA plates for testing pathogens growth, germination efficiency of C. beticola, and growth of T. atroviride on liquid medium for RNA extraction, but does not include the test for T. atroviride in PDA plates (results from Figure 2A). There are some descriptions that can be read in the results section or figure legends but are not well described in this section which make difficult to understand the experiments.

In addition:        

                Lines 210-212: review sentence.                 

                Line 214: circle? – cycle

                Lines 216-217: how many replicates? if n= 2? How many times was the experiment repeated?

                Lines 223-228: it does not describe how T. atroviride was inoculated. Also, describes the growth of T. atroviride on chitosans, N-acetylglucosamine, glucose or glycerol but in lines 320-323, sugar beet samples are mentioned.

                Lines 232-235: these sentences are already mentioned above regarding culture cultivation conditions

                Lines 236-237: T. atroviride and the pathogen were inoculated at the same time? And, all pathogens were evaluated at day 6?

                Line 242: what type of chitosan? How many seeds per concentration of chitosan?

                Calculation formulas for germination indexes are described according to reference number 58 which is not available in any database. Is there any other publication available?

                Sterile conditions were used to cultivate seedlings for RNA extraction (line 253) but not for in vitro cultivation (line 240)? Also, surface-sterilized should be indicated. And for experiment described in lines 253-262, how many seedlings were used per treatment?

                Line 290: Seven days

                Line 294: indicate the control used.

                Line 338: qRT-PCR – indicate RT-qPCR. Please, change it all along the manuscript.

                Indicate in supplementary tables S1 and S2 that primers refer to either T. atroviride or B. vulgaris as well as the description of the gene (cda1 is a predicted chitinase, or cho1 is a chitosanase, etc.)

                Primers used for B. vulgaris: sequences of PALf and PALr do not match with sequence AY453842.1

                PCR cycling conditions for gene expression (RT-PCR and RT-qPCR) are not described.

  1. Results

                Lines 357-369: these sentences should be removed or modified as indicate either introduction or materials and methods and not results. Similarly, for lines 418-425; for lines 434-442; for lines 497-501; for lines 512-519; for lines 556-571; for lines 606-610; for lines 731-742.

                There are sentences such as those in lines 394-395 and 398-399 that refer to Discussion instead of Results. Similarly, for lines 429-431; for lines 452-454; for lines 467-474; for lines 489-496; for lines 507-510; for lines 535-537; for lines 599-602; for lines 611-616; for lines 624-626; for lines 629-634; for lines 644-646; for lines 649-656; for lines 662-666.

                Figures showing statistical differences should be indicated either with an asterisk or any other kind of label. Please indicates the meaning of CA in Figure 2.

                Lines 431-433: these lines refers to Figure 1? If it is so, how to explain results for P. citrinum, or A. versicolor?

                Dual confrontation assays were evaluated after 6 days but why it does not include the control where the pathogen is grown alone (Figure 2C) in order to compared its growth under the absence of Trichoderma on similar conditions. 

                Legend of Figure S2: supplementary table A1?

                Lines 502-503 and Figure S2: the expression of cda4 and cda6 genes are equally induced as cda3

                Materials and Methods describing the results showed in Table 1 and Figure 3 should be more accurate as timing described and measurements are confusing.

                Insect frass incidences are not indicated in Materials and Methods in order to understand how the experiment was setup.

                Figure 6B Leaf disc assay legend is confusing if compared to Materials and Methods description. Please indicate whether disease severity is measured by number of plant discs affected by the disease or the number of spots per disc. n=?

  1. Title

I suggest to look for a more descriptive title that better represent what the study is focusing on, not only the effect on sugar beet but mycoparasitism, priming and bioestimulant.

Author Response

Response to Reviewer 3 Comments

The manuscript titled “The multilateral efficacy of chitosan and Trichoderma  on sugar beet” evaluates the biocontrol efficacy of combining Trichoderma atroviride and chitosan against the Cercospora leaf spot disease, but also as a bioestimulant. Even though the results presented by Kappel et al. are interesting in terms of the few studies researching this topic and the potential of applicability of these formulations, and they are well disccused, how these results are presented makes the reading quite difficult and confusing. In this particular, comments listed below might be related to either the inadequate description of the experimental assay or the lack of description at all. Besides these aspects that need to be addressed for clarity and readability, there are many issues regarding editing references or the use of abbreviations for scientific names of microorganisms (when to use them and where).

General Response: First of all, the authors would like to thank the reviewer and the editors for their efforts, the overall positive evaluation, and their helpful comments, which have enabled us to improve the manuscript where necessary.

However, we hope to convince with the revised version of the manuscript, which should therefore correspond to the JoF guidelines. We agree with the spelling and structure comments and apologize for any errors in clarity of the text. We also apologize for any terminology errors - we have only just started this interdisciplinary study, which may explain the inaccuracy. I apologize for use of incorrect terms.

Since the number of references is exorbitant, we agree with the reviewers' opinions, reducing the inappropriate references and we addedcitations that suit the text better as recommended.

Point 1:

    References format, number or not accurately used.

Authors should carefully review references in order to match the requirements recommended by the journal. As an example, journal articles should follow the following criteria: 1. Author 1, A.B.; Author 2, C.D. Title of the article. Abbreviated Journal Name Year, Volume, page range.”  The first reference listed by authors has the following format:

Badawy MEI, Rabea EI: A Biopolymer Chitosan and Its Derivatives as Promising Antimicrobial Agents against Plant Pathogens and Their Applications in Crop Protection. International Journal of Carbohydrate Chemistry 2011, 2011. All references need to be reviewed and edited.

Badawy, M.E.I., and Rabea, E.I. (2011). A Biopolymer Chitosan and Its Derivatives as Promising Antimicrobial Agents against Plant Pathogens and Their Applications in Crop Protection. International Journal of Carbohydrate Chemistry 2011.

Badawy, Mohamed E. I., and Entsar I. Rabea. "A Biopolymer Chitosan and Its Derivatives as Promising Antimicrobial Agents against Plant Pathogens and Their Applications in Crop Protection." International Journal of Carbohydrate Chemistry 2011 (2011).

Response 1:The References conform now to the JoF style format. Output style MDPI chicago/endnote

Point 2: The number of references seems excessive, for example the 6 references used to indicate the benefits of Trichoderma secondary metabolites (Lines 104-107), or references that are not necessary like reference number 63.

Response 2: We reduced the number of references to 105. Appropriate references have been added and now the proposed publications have been introduced where recommended. All changes are highlighted or can be followed in track change mode.

Point 3: The reference number 18 (Fiorentino et al., 2018) related to foliar applications, does not indicate any experimental assay to this regard. When referring to T. atroviride (lines 82-85), at least two of the references (number 20 and 21) do not evaluated the capacity of this specie against those pathogens or any pathogen at all.

    Introduction

Introduction is excessively extent and could be shorted, for example when referring to the aim of this research study (from line 149 to line 178).

Response 3: We have shortened the introduction as suggested. In addition, redundant passages have been moved to the discusson section.

Point 4:     Line 94: “which”, should be corrected.

Response 4: corrected

Point 5:     Lines 97-100: DAMPs can also trigger defense responses in the plant-Trichoderma

Response 5: rephrased

Point 6: Lines 100-103: this sentence, referring to cerato-platanins of pathogens (references 29 and 30) is included in a paragraph referring to the properties of Trichoderma. There are many articles that describes these proteins like the well-known SM1 elicitor that might fit better that the current sentence. Also, lines 103-104 refer to EPL1 as a continuation of the description of the properties.

Response 6: changed

Point 7: Lines 138-139: reference number 20 is not related to what the sentence is indicating (fungal resistances).

Response 7: changed

    Point 8Line 155: “growing resistance of pests” or plant diseases?

Response 8: changed

    Point 9Line 170-171: CLS abbreviation.

Response 9: corrected

   Point 10: Materials and Methods.

Titles of this section should be more descriptive. As an example, cultivation conditions section (line 205) includes preparation of PDA plates for testing pathogens growth, germination efficiency of C. beticola, and growth of T. atroviride on liquid medium for RNA extraction, but does not include the test for T. atroviride in PDA plates (results from Figure 2A). There are some descriptions that can be read in the results section or figure legends but are not well described in this section which make difficult to understand the experiments.

Response 10: The titles of this subsection are now more descriptive. The experimental setup for T. atroviride in PDA plates has been described accordingly. Missing descriptions have also been added to Material and Methods.

In addition:       

Point 11: Lines 210-212: review sentence.                

Response 11: rephrased

Point 12: Line 214: circle? – cycle

Response 12: We corrected the spellingerror

Point 13: Lines 216-217: how many replicates? if n= 2? How many times was the experiment repeated?

Response 13: We added the information: 2 biological with 2 technical replicates.

Point 14: Lines 223-228: it does not describe how T. atroviride was inoculated. Also, describes the growth of T. atroviride on chitosans, N-acetylglucosamine, glucose or glycerol but in lines 320-323, sugar beet samples are mentioned.

Response 14: We revised the sentences and also stated now that  “1 x 10^6 T. atroviride spores were used to inoculated the medium.” and that the “Mycelium was harvested after 24 h of growth at 28°C”. We also rephrased the sentence in 320-323. The QIAGEN kit was used for both, the indicated T. atroviride samples, but also for the extraction of RNA from B. vulgaris samples from other experiments.

Point 15: Lines 232-235: these sentences are already mentioned above regarding culture cultivation conditions

Response 15: The repeated content has been deleted.

Point 16: Lines 236-237: T. atroviride and the pathogen were inoculated at the same time? And, all pathogens were evaluated at day 6?

Response 16: We added the information: The growth of the hosts B. cinerea, S. sclerotiorum, F. oxysporum was monitored after six days in circadian illumination at 28°C. C. beticola, due to its very slow growth was placed on the plates six days prior to T. atroviride and then measured after another six days of incubation.

Point 17: Line 242: what type of chitosan? How many seeds per concentration of chitosan?

Response 17: corrected:

We added the information: HMW chitosan; 100 seeds per condition. 1 concentration of HMW chitosan (10mg/ml)

Point 18: Calculation formulas for germination indexes are described according to reference number 58 which is not available in any database. Is there any other publication available?

Response 18: A new reference has been provided concerning the content.

Point 19: Sterile conditions were used to cultivate seedlings for RNA extraction (line 253) but not for in vitro cultivation (line 240)? Also, surface-sterilized should be indicated. And for experiment described in lines 253-262, how many seedlings were used per treatment?

Response 19: Yes, only for RNA extraction, where the seedlings were cultured on Phytagel allowing easy contamination due to sucrose supplementation. The seedlings were additionally surface sterilized beforehand as described in MM. We wanted to be sure that these measured effects were solely due to chitosan or T. atrovirid and not to other effects/microorganisms that might inoculate the seed.

However, in order to rule out a negative influence on germination, the germination test was carried out on filter paper without prior surface disinfection.

Surface sterilized is now indicated.

We now indicate that 5 seeds were used per condition for growth in phytagel and that the experiment was repeated twice.

Point 20: Line 290: Seven days

Response 20: corrected

Point 21:Line 294: indicate the control used.

Response 21: all plants were sprayed with C. beticola spores. As control served plants that had been primed only with acidified water as described in line 285 of the original manuscript and were compared to the plants primed with either chitosan or T. atroviride. We rephrased the sentence to clarify this.

Point 22: Line 338: qRT-PCR – indicate RT-qPCR. Please, change it all along the manuscript.

Response 22: corrected

Point 23: Indicate in supplementary tables S1 and S2 that primers refer to either T. atroviride or B. vulgaris as well as the description of the gene (cda1 is a predicted chitinase, or cho1 is a chitosanase, etc.)

Response 23: indicated

Point 24: Primers used for B. vulgaris: sequences of PALf and PALr do not match with sequence AY453842.1

Response 24: We apologize, that the sequence number was confused. The sequence identifier was corrected to BQ585675.1

Point 25: PCR cycling conditions for gene expression (RT-PCR and RT-qPCR) are not described.

Response 25: This has been corrected for RT-PCR. For terms related to RT-qPCR, the citation containing the information has already been given.

     Results

Point 26: Lines 357-369: these sentences should be removed or modified as indicate either introduction or materials and methods and not results. Similarly, for lines 418-425; for lines 434-442; for lines 497-501; for lines 512-519; for lines 556-571; for lines 606-610; for lines 731-742.

Response 26: We removed the content that was either introduction or materials and methods accordingly.

Point 27: There are sentences such as those in lines 394-395 and 398-399 that refer to Discussion instead of Results. Similarly, for lines 429-431; for lines 452-454; for lines 467-474; for lines 489-496; for lines 507-510; for lines 535-537; for lines 599-602; for lines 611-616; for lines 624-626; for lines 629-634; for lines 644-646; for lines 649-656; for lines 662-666.

Response 27: We removed content referring to discussion or transferred important content to the discussion section. We did, however not delete the sentences 489-493, but rephrased them since they were needed to state why the next experiment was performed.

Point 28:Figures showing statistical differences should be indicated either with an asterisk or any other kind of label. Please indicates the meaning of CA in Figure 2.

Response 28: We now included data for statistical significance for the qPCR analysis in the supplementary file tables A3 and A4 and were applicable in the figures themselves. CA in Figure 2 was indicated as confrontation assay.

Point 29: Lines 431-433: these lines refers to Figure 1? If it is so, how to explain results for P. citrinum, or A. versicolor?

Response 29: We rephrased the sentence so that it should become clear now, that this statement only refers to the plant pathogens which are sensitive to chitosan. We conclude that T. atroviride can establish itself in the presence of high concentrations of chitosan with different grades. By contrast, in the plant pathogens, which were sensitive to chitosan, we did not observe an adaption to chitosan even after prolonged incubation times.

Point 30:Dual confrontation assays were evaluated after 6 days but why it does not include the control where the pathogen is grown alone (Figure 2C) in order to compared its growth under the absence of Trichoderma on similar conditions.

Response 30: We did not include photographs of the pathogens alone under all conditions in the original manuscript, to reduce redundant content, since data for growth on chitosans was already presented in figure 1. We include now a supplementary figure showing the reference growth of the pathogens on the chitosans and PDA without confrontation with T. atroviride corresponding to figure 2C.

Point 31: Legend of Figure S2: supplementary table A1?

Response 31: All supplementary table and figure numbers are changed now to “S” instead of “A”.

Point 32: Lines 502-503 and Figure S2: the expression of cda4 and cda6 genes are equally induced as cda3

Response 32: The reviewer is correct that the expression of all three cdas on chitosans is similar, but the expression levels should be compared to the non-induced conditions: glucose or glycerol. Expression is lower or similar in cda4 and cda6 on chitosans compared to control conditions, but for cda3 expression is slightly higher on chitosans than on glucose or glycerol.

Point 33: Materials and Methods describing the results showed in Table 1 and Figure 3 should be more accurate as timing described and measurements are confusing.

Response 33: We rephrased the paragraph in the materials and methods section and hope that the methodology and calculations are comprehensible now.

Point 34: Insect frass incidences are not indicated in Materials and Methods in order to understand how the experiment was setup.

Response 34: The incidence of frass (%) was included in M&M              

Point 35: Figure 6B Leaf disc assay legend is confusing if compared to Materials and Methods description. Please indicate whether disease severity is measured by number of plant discs affected by the disease or the number of spots per disc. n=?

Response 35: In order to present this more clearly, we have rephrased the sentences in the material and methods section and in the figure legend.

Point 35 Title

I suggest to look for a more descriptive title that better represent what the study is focusing on, not only the effect on sugar beet but mycoparasitism, priming and bioestimulant.

Response 35:  We propose another title - but feel that long titles always lose meaning even the titel was descriptive. Suggested is now:

The multilateral efficacy of Trichoderma and Chitosan promotes growth, stimulates the defense response and primes for disease resistance of Beta vulgaris to Cercospora beticola.

Round 2

Reviewer 1 Report

1.Although the manuscript has been revised, it is still too long. The test results were not highly condensed. The discussion was also not highly concise. As this manuscript is not a literature review, no more than 40 references are recommended. 2.In this manuscript, the experimental results presented are still too superficial. More studies on the mechanism of the multiple efficacy of Trichoderma and Chitosan promotes growth and stimulates the defense response should be supplemented. Therefore, the manuscript does not meet the publication requirements. It is suggested to reject the manuscript.

Author Response

Response to reviewer 1

Comments and Suggestions for Authors

Point 1: Although the manuscript has been revised, it is still too long. The test results were not highly condensed. The discussion was also not highly concise. As this manuscript is not a literature review, no more than 40 references are recommended.

Response 1: The citations have again been significantly reduced and the manuscript now contains 64 citations as suggested by Reviewer 1 and the editor and is consistent with other publications within JoF. Only 40 citations in a manuscript type like the current one is not common. The results are condensed and presented in a clear way. The discussion in our opinion is concise and also the reviewers 2 and 3 are satisfied with information and hypothesis provided in the whole manuscript. The experiments are sufficient to draw the presented conclusions. The manuscript contains innovative and important insights for the community.

Point 2: In this manuscript, the experimental results presented are still too superficial. More studies on the mechanism of the multiple efficacy of Trichoderma and Chitosan promotes growth and stimulates the defense response should be supplemented. Therefore, the manuscript does not meet the publication requirements. It is suggested to reject the manuscript.

Response 2:

Superficial experimental design: We believe that primary findings are confused with superficiality. A hypothesis is proposed to test the concept; with later investigation of a complex intracellular mechanism, e.g. “Explore the internal mechanism of chitosan and Trichoderma in sugar beets” as suggested in the first round. What exactly is superficial about a series of in vitro tests followed by in vitro germination tests and an initial plant trial, gene expression studies and numerous germination tests with both agents? The number of methods performed or the type of experiments used?

In a case of a rejection, the reviewer should please endeavor to specify what exactly superficial means. The results or effects have been shown in two or more ways! We are biologists and investigations into the physical/biochemical intracellular mechanism of chitosan are outside the scope.

We also present a first in vitro assay that combines T. atroviride and chitosan, which leads to a synergistic growth reduction of important sugar beet pathogens. The study clearly shows the high potential for a combined use of the two agents and is clearly an innovative approach.

In any case, the conclusions that one could draw from these results are novel and the experiments are not superficial and are indeed, in our opinion, very important for the biocontrol community.

Reviewer 2 Report

The present version has been clearly improved but some typographic things should be adjusted by authors.

- Line 59: … as seed [14],

- Line 119: pests should be changed by pathogens

- Line 133:  … “as potent B. vulgaris biocontrol agents”? This sentence should be improved.

- Line 142: Authors must unify temperatura and time formats along the manuscript. i.e. 28 ºC or 28ºC?, 48h, 72h or 48 h, 72 h? Please, check the entire document.

- Line 162: Author must unify names of microorganism in subtitles (italics or not). i.e. 162, 205, 235, 332, 369, 432, and successively.

- Line 235: B. vulgaris (abbreviation)

- Lines 341-342: LMW and HMW chitosan were …. (these abbreviations were previously indicated)

- Lines 642 and 643: … LMW and HMW chitosan (abbreviations are enough).

- Line 698: … upregulated.

- The section “References” must be carefully checked since different formats were used also in the version 2. When authors refer the journal names (abbreviated or not, …) and the title of articles (capital letters or not).

Author Response

Response to Reviewer 2 Comments/2

Point 1: The present version has been clearly improved but some typographic things should be adjusted by authors.

Response: We would like to thank the reviewer again for his/her efforts, the overall positive evaluation, and the helpful comments, which have enabled us to improve the manuscript where necessary. Finally we hope to convince with the revised version of the manuscript, which should therefore correspond to the JoF guidelines

Point 1: Line 59: … as seed [14],

Response 1: Corrected. Citation [14] Xue et al., 2017 was deleted according to the Editors suggestions also.

Point 1: Line 119: pests should be changed by pathogens

Response1: Corrected.

Point 1: Line 133:  … “as potent B. vulgaris biocontrol agents”? This sentence should be improved.

Response 1: The sentence was rephrased.

Point 2: Line 142: Authors must unify temperature and time formats along the manuscript. i.e. 28 ºC or 28ºC?, 48h, 72h or 48 h, 72 h? Please, check the entire document.

Response 2: The entire document was checked and a unified format XX °C and XX h, etc. was used.

Point 3:  Line 162: Author must unify names of microorganism in subtitles (italics or not). i.e. 162, 205, 235, 332, 369, 432, and successively.

Response 3: The document names of microorganisms in subtitles were checked and all are not italic now.

Point 4: Line 235: B. vulgaris (abbreviation)

Response 4: corrected.

Point 5:  Lines 341-342: LMW and HMW chitosan were …. (These abbreviations were previously indicated)

Response 5: corrected

Point 6: Lines 642 and 643: … LMW and HMW chitosan (abbreviations are enough).

Response 6: corrected

Point 7: Line 698: … upregulated.

Response 7: corrected

Point 8: The section “References” must be carefully checked since different formats were used also in the version 2. When authors refer the journal names (abbreviated or not, …) and the title of articles (capital letters or not).

Response 8: We corrected the reference style in the manuscript and in the supplementary material file.

Reviewer 3 Report

Overall, the manuscript has been improved, however there are still some issues regarding editing that should be addressed before publishing.

  1. Even though bibliography software packages are recommended for preparing the references, these should be double-checked one by one to avoid typing mistakes.

According to JoF Instructions for Authors:

Journal Articles:
1. Author 1, A.B.; Author 2, C.D. Title of the article. Abbreviated Journal Name Year, Volume, page range.

In this particular case, abbreviated journal names are not indicated (most of them), and uppercase letters are used in some titles of the publications (cites 16 and 84). References included in the Supplementary material show similar errors.

  1. There are still some typo errors; few examples are indicated next:

Line 66: “One of these promising biological control agents is the mycoparasite Trichoderma atroviride.”: BCA

Line 760: “Nielsen 1994 showed” [?]

Some symbols (oC or o C, hours, ...) should follow the same criteria.

  1. Regarding point 28. Even though statistical differences have been included for Figure 4 and 5, I referred to the lack of statistical differences in Figure 2B and 2D and Figure 6. They should be indicated similarly to how the results in Figure 3 are presented.
  2. Regarding point 35 (title), authors might be right so the title could be presented as initially intended.

Author Response

Response to Reviewer 3 Comments/2

Overall, the manuscript has been improved, however there are still some issues regarding editing that should be addressed before publishing.

Response: We would like to thank the reviewer for the overall positive evaluation, helpful comments and expert opinion on our work as well as for the detailed feedback and thus efforts he/she made to improve the structure and terminology. We also apologize for bibliography errors in any cases.

We have checked the entire manuscript and corrected where suggested. We finally hope, that the improvements made will led to a version which can be published in JoF.

Point 1: Even though bibliography software packages are recommended for preparing the references, these should be double-checked one by one to avoid typing mistakes.

According to JoF Instructions for Authors:

Journal Articles:
1. Author 1, A.B.; Author 2, C.D. Title of the article. Abbreviated Journal Name Year, Volume, page range.

In this particular case, abbreviated journal names are not indicated (most of them), and uppercase letters are used in some titles of the publications (cites 16 and 84). References included in the Supplementary material show similar errors.

Response 1: We corrected reference style in the manuscript and in the supplementary material file.

Point 2: There are still some typo errors; few examples are indicated next:

Line 66: “One of these promising biological control agents is the mycoparasite Trichoderma atroviride.”: BCA

Response 2: We reviewed the whole manuscript again for biological control agent or biocontrol agent and replaced all with BCA after the term was mentioned for the first time in the manuscript.

Point 3: Line 760: “Nielsen 1994 showed” [?]

Respones 3: The citation was moved up from line 762 directly after “Nielsen 1994”

Point 4. Some symbols (oC or o C, hours, ...) should follow the same criteria.

Response 4: The entire document was checked and a unified format XX °C and XX h or XX d was used.

Point 5. Regarding point 28. Even though statistical differences have been included for Figure 4 and 5, I referred to the lack of statistical differences in Figure 2B and 2D and Figure 6. They should be indicated similarly to how the results in Figure 3 are presented.

Response 5/6: The statistical differences for Figure 6C were included.

For Figure 6B we did not perform statistics, since only the number of discs according to their categories are presented. The leaf-discs were derived from 2 plants per condition, as stated in the methods.

We did not perform statistics for Figures 2B and 2D because data derived from 2 experiments (n=2, technical replicates) with 2 biological replicates (2 plates per condition) each. The data were highly reproducible (given by the small standard deviations). We consider the data to be compliant with the rules for biological experiments. We included a sentence stating that standard deviations are given in the figure legend and the number of experiments and biological replicates were already given in the previous versions in the Methods section

Point 7: Regarding point 35 (title), authors might be right so the title could be presented as initially intended.

Response 7: The title is now written as originally